# Test Time Scaling for Neural Processes

**Hyungi Lee**
Kookmin University
lhk2708@kookmin.ac.kr

**Moonseok Choi**
KAIST
ms.choi@kaist.ac.kr

**Hyunsu Kim**
KAIST
kim.hyunsu@kaist.ac.kr

**Kyunghyun Cho**
New York University&Genetech
kyunghyun.cho@nyu.edu

**Rajesh Ranganath**
New York University
rajeshr@cims.nyu.edu

**Juho Lee**
KAIST
juholee@kaist.ac.kr

## Abstract

Uncertainty-aware meta-learning aims not only for rapid adaptation to new tasks but also for reliable uncertainty estimation under limited supervision. Neural Processes (NPs) offer a flexible solution by learning implicit stochastic processes directly from data, often using a global latent variable to capture functional uncertainty. However, we empirically find that variational posteriors for this global latent variable are frequently miscalibrated, limiting both predictive accuracy and the reliability of uncertainty estimates. To address this issue, we propose Test Time Scaling for Neural Processes (TTSNPs), a sequential inference framework based on Sequential Monte Carlo Sampler (SMCS) that refines latent samples at test time without modifying the pre-trained NP model. TTSNPs iteratively transform variational samples into better approximations of the true posterior using neural transition kernels, significantly improving both prediction quality and uncertainty calibration. This makes NPs more robust and trustworthy, extending applicability to various scenarios requiring well-calibrated uncertainty estimates.

## 1 Introduction

In many real-world scenarios, tasks naturally vary in several aspects, such as the number of available data points and the degree of correlation between training and evaluation data. Despite the diversity, a common challenge remains: each task typically offers only limited supervision, which makes it difficult to train models that generalize reliably within individual tasks [2, 27]. To address this issue, meta-learning [50, 15] has emerged as a powerful paradigm. While meta-learning [15, 25, 22] has demonstrated significant success in improving adaptability in various fields via extracting transferable knowledge, reliable uncertainty estimation along with accurate predictions becomes important, as it allows models to express appropriate levels of confidence in their predictions and make trustworthy decisions given limited information. To this end, uncertainty-aware meta-learning [48, 1, 33, 49] has been proposed to integrate uncertainty quantification directly into meta-training and evaluation processes, promoting both rapid adaptation and calibrated prediction.

Among various frameworks, Neural Processes [NPs; 18, 19] have emerged as a flexible and data-driven approach for uncertainty-aware meta-learning. By training parametric neural networks to maximize predictive likelihoods across meta-task datasets, NPs can capture the data-generating mechanisms underlying diverse tasks without relying on explicit prior assumptions. As a result,

39th Conference on Neural Information Processing Systems (NeurIPS 2025).

they offer a powerful balance between flexibility and uncertainty estimation, positioning them as a foundational model for uncertainty-aware meta-learning [33]. A common approach to modeling uncertainty in NPs involves introducing a global latent variable, which encodes functional uncertainty or epistemic uncertainty across tasks.

However, in this work, we empirically demonstrate that previous approaches on NP models [19, 28, 33] leveraging a global latent variable to model functional uncertainty often suffer from miscalibrated variational posteriors for the global latent variable. Moreover, we find that such miscalibration issues consistently arise regardless of the specific loss functions or model structure used during the training of NPs. This miscalibration presents a fundamental challenge for NPs, where the global latent variable serves as the core mechanism for conveying contextual information to target predictions. When the variational posterior fails to faithfully reflect the uncertainty in the context set, the resulting latent representations may become either overly deterministic or excessively diffuse. Consequently, the model's predictions on target points can suffer from poor calibration and reduced expressiveness, particularly when the target inputs deviate from those observed in the context set. These limitations undermine the generalization ability and reliability of NPs, especially in scenarios where accurate uncertainty estimation is crucial.

To overcome this limitation, we propose a principled sequential inference framework based on Sequential Monte Carlo Samplers (SMCS) [11, 10], designed to refine latent variable samples at test time without modifying the pre-trained NP model. Specifically, our approach, Test Time Scaling for Neural Processes (TTSNPs), iteratively transforms latent samples from the variational posterior into more accurate approximations of the true posterior using SMCS, with the following two key components: (1) learned neural network transition kernels, and (2) learned intermediate distributions, achieved by generating pseudo context points via a neural network. This refinement improves predictive performance and enhances uncertainty calibration, enabling NPs to produce more reliable and trustworthy predictions. As a result, TTSNPs significantly extend the applicability of NPs to various tasks that demand robust and well-calibrated uncertainty estimation.

## 2 Preliminaries

### 2.1 Problem Setup

In typical NP settings, the dimensionality of datasets is assumed to be fixed [18, 19, 28, 32]. In contrast, following the direction explored in Lee et al. [33], we consider a more general setting in which the data for each task may vary in dimensionality. Specifically, we assume potentially infinite collection of tasks $\{\tau_j\}_{j \in \mathbb{N}}$, each drawn i.i.d. from a task distribution $p_\tau(\tau)$. Every task $\tau_j$ uses an input space $\mathcal{X}_{d_x(j)} \subseteq \mathbb{R}^{d_x(j)}$ and output space $\mathcal{Y}_{d_y(j)} \subseteq \mathbb{R}^{d_y(j)}$, where $d_x,\ d_y \colon \mathbb{N} \to \mathbb{N}$ specify each task's input-output dimensionality. Concretely, $\tau_j$ provides a dataset $\mathcal{D}_j = \{\mathbf{d}_{j,k}\}_{k=1}^{n_j}$ in which each $\mathbf{d}_{j,k} = (\mathbf{x}_{j,k}, \mathbf{y}_{j,k})$ belongs to $\mathcal{X}_{d_x(j)} \times \mathcal{Y}_{d_y(j)}$. A subset $c_j \subsetneq [n_j]$ of indices defines the *context set* $\mathcal{D}_{j,c} := \{\mathbf{d}_{j,k}\}_{k \in c_j}$, while the complement $t_j = [n_j] \setminus c_j$ gives the *target set* $\mathcal{D}_{j,t}$. Each dataset $\mathcal{D}_j$ is considered i.i.d. from some unknown function $f_j \colon \mathcal{X}_{d_x(j)} \to \mathcal{Y}_{d_y(j)}$. Meta-learning aims to learn a *shared* representation capturing how $\mathbf{x}_{j,k}$ and $\mathbf{y}_{j,k}$ are related, using $\mathcal{D}_{j,c}$ (training) and $\mathcal{D}_{j,t}$ (validation) across many tasks.

### 2.2 Neural Processes

NPs address the meta-learning problem by defining a distribution over functions $f_j$ conditioned on each context set $\mathcal{D}_{j,c}$. The predictive distribution can be viewed as follows [33]:

$$p(\mathbf{Y}_j \mid \mathbf{X}_j, \mathcal{D}_{j,c}) = \int \Big[ \prod_{k \in [n_j]} p(\mathbf{y}_{j,k} \mid f_j, \mathbf{x}_{j,k}) \Big] p(f_j \mid \mathcal{D}_{j,c}) \, \mathrm{d}f_j, \tag{1}$$

where $\mathbf{X}_j$ and $\mathbf{Y}_j$ collect all inputs $\{\mathbf{x}_{j,k}\}$ and outputs $\{\mathbf{y}_{j,k}\}$. When adopting a Gaussian likelihood for $p(\mathbf{y}|f, \mathbf{x})$ and introducing a latent variable $r_j \in \mathbb{R}^{d_r}$ that parameterizes $f_j$, one might write

$$p(\mathbf{Y}_j \mid \mathbf{X}_j, \mathcal{D}_{j,c}) = \int \prod_{k \in [n_j]} \mathcal{N}\big(\mathbf{y}_{j,k} \mid \mu_{r_j}(\mathbf{x}_{j,k}), \mathrm{diag}(\sigma_{\mathbf{r}_j}^2(\mathbf{x}_{j,k}))\big) \, q(\mathbf{r}_j \mid \mathcal{D}_{j,c}; \boldsymbol{\phi}) \, \mathrm{d}\mathbf{r}_j, \tag{2}$$

where $\phi$ is the parameter set of the NPs encoder module. NP variants differ primarily in how they encode $\mathcal{D}_{j,c}$ into $r_j$. Conditional NPs (CNPs) [18, 21, 48] use a deterministic encoder that maps $\mathcal{D}_{j,c}$ to a point estimate $\bar{\mathbf{r}}_j$. Latent NPs [19, 17, 32] instead model uncertainty by placing a variational Gaussian over $\mathbf{r}_j$. Both approaches generate predictive means and variances via a decoder:

$$(\mu_{r_j}, \sigma_{r_j}^2) = f_{\mathrm{dec}}(\mathbf{x}_{j,k}, r_j; \boldsymbol{\psi}), \tag{3}$$

where $\boldsymbol{\psi}$ is the parameter set of the NPs decoder module. CNPs maximize the predictive log-likelihood over meta-training tasks, while latent NPs optimize the predictive log-likelihood or an Evidence Lower BOund (ELBO) to balance data fit and Kullback-Leibler (KL) regularization. Here, ELBO is usually approximated by the following form:

$$\mathbb{E}_{\tau_j}[\log p(\mathbf{Y}_j \mid \mathbf{X}_j, \mathcal{D}_{j,c})] \geq \mathbb{E}_{\tau_j}\Big[\sum_{k \in [n_j]} \mathbb{E}_{q(r_j \mid \mathcal{D}_j)}\big[\log \mathcal{N}_{j,k}\big] - \mathrm{KL}\big[q(r_j \mid \mathcal{D}_j) \,\big\|\, q(r_j \mid \mathcal{D}_{j,c})\big]\Big],$$
$$\tag{4}$$

where $\mathcal{N}_{j,k}$ is shorthand for $\mathcal{N}\big(\mathbf{y}_{j,k} \mid \mu_{r_j}(\mathbf{x}_{j,k}), \mathrm{diag}(\sigma_{r_j}^2(\mathbf{x}_{j,k}))\big)$. In this paper, we are focusing on latent NPs, where we perform test-time scaling to sample from the latent variable more accurately.

### 2.3 Sequential Monte Carlo Samplers

In this section, we briefly review Sequential Monte Carlo Samplers (SMCS) [11], which form the foundational framework for our test-time scaling approach. For a more comprehensive overview, we refer the reader to Del Moral et al. [11], Dai et al. [10].

Let $\pi(x) := \gamma(x)/Z$ denote a target distribution, where the unnormalized density $\gamma(x)$ is known pointwise but the normalization constant $Z$ is unknown. SMCS aims to approximate expectations with respect to $\pi$, such as $\mathbb{E}\pi[f]$ for a test function $f$, while simultaneously providing an estimate of $Z$. Rather than sampling directly from the target $\pi$, SMCS constructs a sequence of intermediate distributions $\{\pi_t\}_{t=0}^T$, beginning with a tractable initial distribution $\pi_0$ and gradually transporting it toward the target $\pi_T := \pi$. These intermediate distributions are designed to interpolate between $\pi_0$ and $\pi$, typically becoming increasingly complex as $t$ increases. Since the intermediate distributions are generally not analytically tractable, SMCS employs sequential importance sampling to adjust the samples accordingly.

In detail, let $\gamma_t$ be the unnormalized density of $\pi_t$, i.e., $\pi_t(x) = \gamma_t(x)/Z_t$ for $t = 0, \ldots, T$ (note that $Z_0 = 1$ as $\pi_0$ is assumed to be fully known). SMCS introduce *forward transition kernels* $\{F_t(x_t \mid x_{t-1})\}_{t=1}^T$ which propagates a sample $x_{t-1}$ to $x_t$. Starting with $x_0 \sim \pi_0$, a sample path $x_{0:T} := (x_0, \ldots, x_T)$ is generated from the joint proposal with density,

$$Q(x_{0:T}) := \pi_0(x_0) \prod_{t=1}^T F_t(x_t \mid x_{t-1}). \tag{5}$$

In principle, one would like to use the marginal $\int Q(x_{0:T}) \mathrm{d}x_{0:T-1}$ as a proposal density for $\pi$, but this is generally intractable. Instead, SMCS augments the target to the path space $x_{0:T}$ with a sequence of *backward transitions* $\{B_{t-1}(x_{t-1} \mid x_t)\}_{t=1}^T$ and defines the unnormalized path density

$$\Pi(x_{0:T}) := \frac{\Gamma(x_{0:T})}{Z} \text{ where } \Gamma(x_{0:T}) = \gamma(x_T) \prod_{t=1}^T B_{t-1}(x_{t-1}|x_t). \tag{6}$$

A sample path $x_{0:T}$ drawn from $Q$ is then corrected with the importance weights,

$$w_T(x_{0:T}) := \frac{\Gamma(x_{0:T})}{Q(x_{0:T})} = \frac{\gamma(x_T) \prod_{t=1}^T B_{t-1}(x_{t-1} \mid x_t)}{\pi_0(x_0) \prod_{t=1}^T F_t(x_t \mid x_{t-1})}. \tag{7}$$

Given the forward and backward kernels, SMCS starts by drawing $N$ i.i.d. *particles* $\{x_0^i\}_{i=1}^N$ from $\pi_0$. At each iteration, the particles are sequentially extended by drawing samples from the forward kernel $F_t$, and the extended particles are re-weighted through the importance weights, which are computed in recursive fashion as follows:

$$w_0(x_0^i) = 1, \quad w_t(x_{0:t}^i) = w_{t-1}(x_{0:t-1}^i) \times \frac{\gamma_t(x_t^i) B_{t-1}(x_{t-1}^i \mid x_t^i)}{\gamma_{t-1}(x_{t-1}^i) F_t(x_t^i \mid x_{t-1}^i)} \text{ for } i = 1, \ldots, N. \tag{8}$$

After completing the sequential update up to the step $T$, we have an estimator for the expectation $\mathbb{E}_\pi[f]$ and the normalization constant $Z$,

$$\mathbb{E}_\pi[f] \approx \sum_{i=1}^{N} \frac{w_T(x_{0:T}^i)}{\sum_{j=1}^{N} w_T(x_{0:T}^j)} f(x_T^i), \quad Z \approx \frac{1}{N} \sum_{i=1}^{n} w_T(x_{0:T}^i). \tag{9}$$

**Learning for SMCS.**  Since both the intermediate distributions and the forward/backward kernels are crucial design choices in SMCS, it is beneficial to learn them when possible. Learning for SMCS can be formulated as a variational inference problem, where the ELBO provides a lower bound on the marginal likelihood defined by the SMCS procedure. A common approach is to begin with unadjusted Langevin diffusions and modify the drift term (involving the score functions) to maximize the ELBO [12, 20, 53, 7]. Among these, Chen et al. [7] introduced an optimal control framework for tuning a continuous-time extension of SMCS with resampling, which forms the foundation of our proposed algorithm. Refer to Appendix B to see additional explanations for SMCS.

## 3 Test Time Scaling for Neural Processes

### 3.1 Observation

As discussed in § 2.2, latent NP models are typically trained by maximizing the predictive log-likelihood or ELBO. Ideally, this results in a variational posterior $q(\mathbf{r}|\mathcal{D}_c)$ that approximates the true posterior $p(\mathbf{r}|\mathcal{D}_c) \propto \prod_{(x',y')\in\mathcal{D}_c} p(y'|x',\mathbf{r})p(\mathbf{r})$. However, due to training limitations and overfitting to specific context distributions, the learned posterior often deviates from the true posterior, especially at test time.

To validate this issue, we conducted experiments using a simple NP model [19] trained on 1D GP data with an RBF kernel. The same trained model was evaluated on unseen test datasets sampled under identical conditions. We compared the target marginal likelihoods using latent samples drawn from the variational posterior, as well as from importance sampling [IS; 52], Hamiltonian Monte

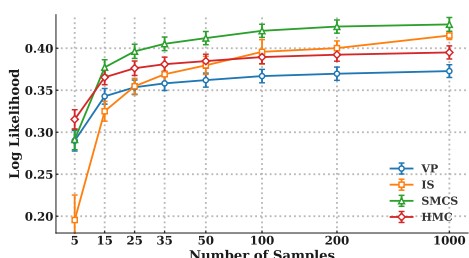

Figure 1: Marginal log likelihood comparison across variational posterior (VP), IS, HMC, and SMCS using the NP model [19]. The x-axis indicates the number of latent samples used for estimation.

Carlo [HMC; 46], and SMCS [11] with Unadjusted Langevin Algorithm (ULA) transitions. Since all methods use the same encoder and decoder, differences in log-likelihood stem solely from the quality of latent samples. Fig. 1 shows that the variational posterior underperforms compared to exact inference methods, rapidly saturating after 15–25 samples. This highlights its lack of diversity and poor calibration, indicating a failure to capture epistemic uncertainty. Based on these findings, we introduce a test-time scaling method using SMCS in the next section. We choose SMCS over other samplers due to its strong sample efficiency and fast convergence in high-dimensional regimes [42, 36, 62], which is also reflected in our empirical results.

### 3.2 Test Time Scaling with Sequential Monte Carlo Samplers

As mentioned in § 3.1, even after being trained on large-scale datasets, the latent encoder in latent NP models often fails to fully capture functional uncertainty—particularly when faced with structured or out-of-distribution test data, such as those arising from unseen kernels. To mitigate this issue, we propose a test-time scaling approach that enhances uncertainty estimation without requiring any modification to the model's pre-trained parameters. This design choice allows us to retain the representational power learned from the large-scale training data while refining the posterior inference process during evaluation.

Our key idea is to resample the global latent variable $\mathbf{r}$, which governs functional uncertainty in latent NP variants. Since the quality of uncertainty estimation is directly influenced by how well this latent distribution aligns with the test context, we aim to improve it via a principled sampling-based method. Specifically, we adopt the SMCS framework [11, 7] to transform an initial set of

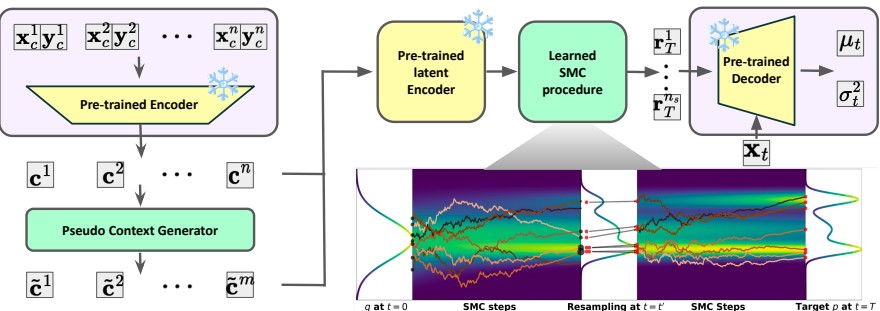

Figure 2: Schematic for TTSNP. TTSNP refines the latent variables from the variational posterior into more accurate samples from the true posterior using the learned transition kernels.

samples drawn from the variational distribution $q(\mathbf{r} \mid D_c)$ into a set that better approximates the true posterior. To implement this, we first construct a sequence of intermediate distributions that gradually interpolate between the initial variational posterior and the target posterior. Then, particle transitions are guided by a parameterized forward kernel and backward kernel, both modeled using a neural network. The forward kernel nudges samples toward high-probability regions of the target distribution, while the backward kernel helps estimate gradients and improve learning stability during training. We also define an SMCS-based training objective derived from KL divergence [41, 7] and the marginal log-likelihood, which facilitates the learning of efficient and effective transitions for posterior refinement. To define an SMCS procedure, we must specify the initial distribution, intermediate distributions, and transition kernels. We detail our choices for each below.

**Initial distribution.** While the initial distribution is typically chosen to be simple, in our case with a pre-trained NP model, the variational posterior $q(\mathbf{r} \mid \mathcal{D}_c) = \mathcal{N}(\mathbf{r} \mid \mu_q, \sigma_q^2 I)$ serves as an effective starting point. Although it may not fully capture the uncertainty of the true posterior, it provides an informative and computationally tractable initialization that can be further refined through SMCS.

**Intermediate distributions.** We construct intermediate distributions via a geometric annealing path, enriched with a learned component. Specifically, for $t = 1, \ldots, T$, we define

$$\pi_t(\mathbf{r}) \propto p(\mathbf{r} \mid \mathcal{D}_c)^{\beta_t} \tilde{q}(\mathbf{r} \mid \mathcal{D}_c)^{1-\beta_t} \cdot \tilde{q}(\mathbf{r} \mid \mathcal{D}_c \cup \mathcal{D}_p)^{1-\beta_t}, \quad 0 < \beta_1 < \cdots < \beta_T = 1. \quad (10)$$

Here, the first term $p(\mathbf{r} \mid \mathcal{D}_c)$ is the true posterior. The second term includes a recalibrated variational posterior $\tilde{q}(\mathbf{r} \mid \mathcal{D}_c) := \mathcal{N}(\mathbf{r} \mid \mu_q, \sigma^2 I)$, retaining the original mean $\mu_q$ but using a fixed variance $\sigma^2$. Empirically, setting $\sigma^2 = 1$ provides stable and robust performance. Together, these two factors form a slight modification of the standard geometric annealing path without requiring additional learning. The third factor introduces our core contribution, where the additional factor is defined as a variational posterior conditioning on the context appended with a *pseudo context set* $\mathcal{D}_p$—a synthetic future context predicted from the current estimate. Concretely, a permutation-invariant neural network $h(\mathcal{D}_c, \varepsilon)$ maps the original context $\mathcal{D}_c$ and noise $\varepsilon \sim p(\varepsilon)$ to a pseudo context $\mathcal{D}_p$, defining an implicit generative model for the future data. Our design is inspired by Martingale posteriors [16, 32], where the predicted future data drive posterior uncertainty. In our case, the pseudo context serves as a form of "lookahead", encouraging the model to maintain hypotheses that remain robust under plausible extensions of the observed context. This mechanism mitigates the overconfidence often observed in amortized inference by injecting uncertainty from possible yet-unseen data. Analogous to "chain-of-thought" reasoning in language models, the pseudo context enables the model to simulate and prepare for potential future scenarios, thereby enriching the intermediate distributions with anticipatory structure and improving posterior coverage. For implementation, we adopt the induced self-attention block [35] for $h$.

**Forward and backward kernels.** Given the intermediate distribution based on pseudo contexts, we define forward and backward kernels based on ULA. Following Chen et al. [7], we formulate our SMCS procedure as a time-discretizations of the continuous-time dynamics governed by the following Stochastic Differential Equation (SDE) evolving over $\tau \in [0, 1]$,

$$\mathrm{d}\mathbf{r}_\tau^u = u(\mathbf{r}_\tau^u, \tau)\mathrm{d}\tau + \sigma \mathrm{d}W_\tau, \quad \mathbf{r}_0^u \sim \pi_0, \quad (11)$$

where $u : \mathbb{R}^{d_r} \times [0, 1] \to \mathbb{R}^{d_r}$ is a drift, $\sigma$ is a diffusion coefficient, and $W_\tau$ is a standard Brownian motion. In parallel, we define the reverse-time SDE,

$$\mathrm{d}\mathbf{s}_\tau^v = v(\mathbf{s}_\tau^v, \tau)\mathrm{d}\tau + \sigma \bar{\mathrm{d}}W_\tau, \quad \mathbf{s}_1^v \sim \pi, \quad (12)$$

with a drift $v$, diffusion $\sigma$, and $\mathrm{d}\bar{W}_\tau$ denoting time-reversed Brownian motion. We expect these two SDES to be time-reversals of each other, and their time-marginals at $\tau_t := t/T$ for $t = 0, \ldots, T$ match the intermediate distributions we set, that is,

$$\mathbb{P}^u_{\tau_t} = \mathbb{P}^v_{\tau_t} = \pi_t \text{ for } t = 0, \ldots, T, \tag{13}$$

where $\mathbb{P}^u_{\tau_t}$ and $\mathbb{P}^v_{\tau_t}$ denote time-marginals of the forward and reverse SDEs at time $\tau_t$. There are potentially infinitely many choices of the pair $(u, v)$, but they must satisfy Nelson's identity [47],

$$u - v = \sigma^2 \nabla \log \pi_\tau \text{ for } \tau \in [0, 1]. \tag{14}$$

Based on this, we choose the structural forms of $u$ and $v$ at $\{\tau_t\}_{t=0}^T$ as

$$\begin{aligned} u(\mathbf{r}_t, \tau_t) &:= \frac{\sigma^2}{2} \left( \nabla \log \tilde{\pi}_t(\mathbf{r}_t) + (1 - \beta_t) \mathrm{NN}(\mathbf{r}_t, t, \mathcal{D}_c, \mathcal{D}_p) \right), \\ v(\mathbf{r}_t, \tau_t) &:= -\frac{\sigma^2}{2} \left( \nabla \log \tilde{\pi}_t(\mathbf{r}_t) + (1 - \beta_t) \mathrm{NN}(\mathbf{r}_t, t, \mathcal{D}_c, \mathcal{D}_p) \right), \end{aligned} \tag{15}$$

where $\mathrm{NN}(\cdot)$ is a neural network that takes as input the state $\mathbf{r}_t$, the step index $t$, and the representations of the context $\mathcal{D}_c \cup \mathcal{D}_p$ computed from a pre-trained encoder. If we were to directly use the score function of $\pi_t$, $\mathrm{NN}$ could simply output the Gaussian score. However, we instead wrap the inputs with a neural net for added flexibility. Note that $\mathrm{NN}$ encompasses this score function as a special case, ensuring compatibility with Nelson's identity. Based on the choice of the drifts, we set the forward and backward transitions for SMCS as Euler-discretizations of the SDEs at $\{\tau_t\}_{t=0}^T$.

$$\begin{aligned} F_t(\mathbf{r}_t \mid \mathbf{r}_{t-1}) &= \mathcal{N}(\mathbf{r}_t \mid \mathbf{r}_{t-1} + h_t u(\mathbf{r}_{t-1}, \tau_{t-1}), 2h_t I), \\ B_{t-1}(\mathbf{r}_{t-1} \mid \mathbf{r}_t) &= \mathcal{N}(\mathbf{r}_{t-1} \mid \mathbf{r}_t + h_t v(\mathbf{r}_t, \tau_t), 2h_t I). \end{aligned} \tag{16}$$

While it is in principle possible to model the entire drifts using a neural network, recent empirical studies [24] indicate that approaches without explicit score functions often fail to work for even simple target distributions. By combining score-based gradients with neural proposals, our method stabilizes particle transport and enables efficient and expressive test-time inference. Refer to Fig. 2 to see the overall model structure and procedure of TTSNP. And, see Appendix B for details on the neural network architecture and modeling choices.

## 3.3 Training Objective

To ensure effective sampling with the SMCS framework, we must carefully design both the training objective and the data used to train $F_t$ and $B_t$. If not properly designed, the importance weights may become overly concentrated on a few samples, leading to a severe particle degeneracy, which can negatively impact the diversity and effectiveness of the sampling process. To promote well-balanced importance weights and obtain high-quality posterior samples, along with the resampling, we employ two complementary loss terms: (1) the Kullback-Leibler (KL) divergence loss between two path measures, denoted as $\mathcal{L}_{\mathrm{KL}}$, and (2) the log-likelihood maximization loss, denoted as $\mathcal{L}_{\mathrm{LL}}$.

**KL Divergence Loss.** The KL divergence loss, $\mathcal{L}_{\mathrm{path}}$, is formulated using two path measures, $\mathbb{P}^u$ and $\mathbb{P}^v$, which correspond to the forward and backward SDEs, respectively. Intuitively, the forward path measure $\mathbb{P}^u$ can be interpreted as the joint distribution $Q(\mathbf{r}^u_{0:T})$ in the limit as $T \to \infty$ [5]. If this divergence is minimized to zero, it implies that the forward and backward transitions are perfectly time-reversible, thereby ensuring ideal sampling [7]. Consequently, training the transition kernels with this objective directly enhances the accuracy and quality of samples generated by the SMCS procedure. Then, given the sequential nature of our approach, the KL divergence is evaluated independently on each subinterval $[\tau_{t-1}, \tau_t]$, yielding:

$$\mathcal{L}_{\mathrm{KL}} = \mathbb{E}_{\mathcal{D}_p} \left[ \sum_{t=1}^T D_{\mathrm{KL}} \left[ \mathbb{P}^u_{[\tau_{t-1}, \tau_t]} || \mathbb{P}^v_{[\tau_{t-1}, \tau_t]} \right] \right], \tag{17}$$

where KL divergence is averaged by pseudo context due to the randomness $\mathcal{D}_p$. This formulation can be rewritten as a simplified objective using importance weights, following Chen et al. [7]. For detailed definitions and the full loss derivation, refer to Appendix B.

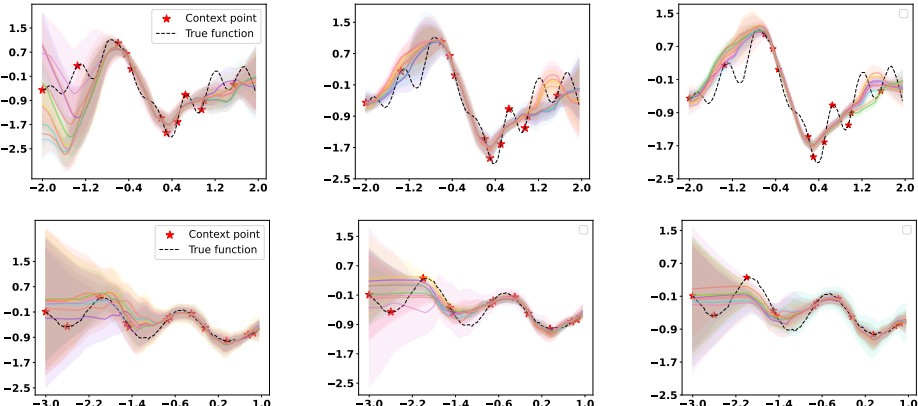

Figure 3: Visualization of posterior samples. (left) Inference results using latent samples drawn from a variational posterior constructed solely from pseudo representations without access to the true context points; (middle) inference results using latent samples refined via TTSNP's SMCS procedure; and (right) inference results from the Fine-tune baseline.

**Marginal Log-likelihood Loss.** In addition to the KL divergence loss, we introduce a log-likelihood maximization loss, $\mathcal{L}_{LL}$, which encourages the generated samples to capture diverse modes of the target distribution. This objective is designed to enhance the representation of epistemic uncertainty while reducing the number of samples required for reliable estimation. The loss is defined as $\mathcal{L}_{LL} = \sum_{(x,y)\in\mathcal{D}} \log \frac{1}{N} \sum_{i=1}^{N} \mathcal{N}(y|x, \mathbf{r}_T^i, \mathcal{D}_c)$, where $\mathbf{r}_T^i$ denotes samples obtained from the final step of the SMCS procedure. Thus our final training objective $\mathcal{L}$ becomes $\mathcal{L} := \mathcal{L}_{KL} + \mathcal{L}_{LL}$.

## 4 Experiments

In this section, we conduct a series of experiments to empirically validate the effectiveness of TTSNPs across a variety of settings, with a particular focus on regression tasks. We utilize two representative models from the latent NP family: simple NP [19], which is the earliest model in this line of work, and DANP [33], a recent model that has demonstrated strong performance and broad applicability across various tasks. Since our TTSNP is designed to be compatible with existing model architectures, we consider the following three baseline approaches: 1) *Pre-train*: a model that has been pre-trained on the training dataset is directly used for inference on the test dataset in a zero-shot manner, relying solely on the pre-trained latent variational posterior without any additional training; 2) *Fine-tune*: the latent path is further adapted for inference by training the transition kernel using the available data at test time; 3) *SMCS*: the samples from variational posterior transported by SMCS with the ULA transition kernels which does not require training. Unless otherwise specified, we fix the number of latent variable samples to 50 across both our method and all baselines to ensure fair comparison. In every result table, the best-performing metric is highlighted in **bold**, and all results are averaged over five different random seeds represented with 1-sigma error bars. In the experiment tables, gray box ■ denotes that the evaluation dataset is the same as the training dataset (seen), while yellow box ■ indicates a different evaluation dataset (unseen) for the TTSNP.

### 4.1 Simple NP

**Sample Quality.** We first conducted an experiment to demonstrate that our method, TTSNP, effectively and efficiently transforms samples drawn from the variational posterior to be better aligned with the true posterior. To this end, we pre-trained an NP model on one-dimensional GP data with RBF kernels. Using additional data from the same kernel, we trained both the 'Fine-tune' baseline and our TTSNP method. We then performed inference on an evaluation set and compared the quality of posterior samples produced by each method. Table 1 clearly demonstrates that TTSNP generates higher-quality latent variables compared to other baselines, *even with the same number of samples*, resulting in improved log-likelihoods for both context and target points. This result shows that our

Table 1: Log-likelihood results for Sample Quality (left) and Matern to RQ (right) scenarios.

| Model | RBF | |
|---|---|---|
| | context | target |
| Pre-train | 0.791 ±0.003 | 0.352 ±0.004 |
| Fine-tune | 0.833 ±0.005 | 0.382 ±0.010 |
| SMCS | 0.868 ±0.001 | 0.405 ±0.001 |
| TTSNP (ours) | **0.893** ±0.001 | **0.430** ±0.001 |

| Model | Matern | | RQ | |
|---|---|---|---|---|
| | context | target | context | target |
| Pre-train | 0.654 ±0.008 | 0.150 ±0.010 | 1.045 ±0.005 | 0.711 ±0.005 |
| Fine-tune | 1.650 ±0.005 | 0.178 ±0.008 | 1.094 ±0.001 | 0.718 ±0.002 |
| SMCS | **0.757** ±0.004 | 0.186 ±0.006 | **1.122** ±0.005 | 0.735 ±0.002 |
| TTSNP (ours) | 0.753 ±0.004 | **0.219** ±0.006 | **1.122** ±0.001 | **0.755** ±0.006 |

learned intermediate distributions and transition kernels generalize effectively not only to additional training data but also to the evaluation data set.

**Analysis on generated pseudo context**   Furthermore, Fig. 3 illustrates how TTSNP benefits from the generated pseudo representations to obtain improved latent posterior samples. Specifically, the left, middle, and right subfigures in Fig. 3 respectively show: (left) inference results using latent samples drawn from a variational posterior constructed solely from pseudo representations without access to true context points, (middle) inference using latent samples refined via TTSNP's SMCS procedure, and (right) inference results from the Fine-tune baseline. As observed in the left figure, latent samples drawn from the variational posterior built on pseudo representations produce predictions and uncertainty estimates that differ from those of the original pre-trained variational poste-

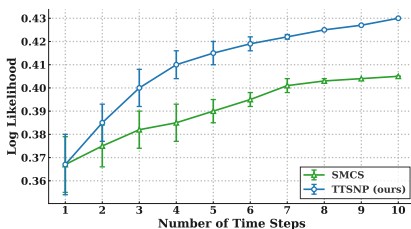

Figure 4: Marginal log likelihood comparison between SMCS and TTSNP using NP model. The x-axis indicates the number of time steps.

rior. This alternative perspective, introduced by the pseudo representations, becomes part of the intermediate distributions used in TTSNP. As a result, during the transition toward the true posterior, such diverse possibilities are continually integrated—allowing TTSNP to capture a broader range of posterior modes more sample-efficiently than standard SMCS.

**Analysis on trade off between inference cost and the performance**   We conducted experiments to analyze the trade-off between inference cost and performance at test time. Specifically, we compared performance across different numbers of SMC steps, rather than using the default setting of $T = 10$ from previous experiments. As shown in Fig. 4, both SMCS and TTSNP exhibit improved performance as the number of inference steps increases. Notably, TTSNP achieves faster performance gains compared to SMCS, demonstrating its greater sample efficiency during inference. This behavior is similar to the trend observed in LLMs, where increasing inference cost—such as through more sampling or decoding steps—often leads to improved model performance.

**Matern to RQ.**   In this scenario, we assume a pre-trained NP model that has been trained on GP data generated using the RBF kernel. We aim to perform inference on GP data generated with either a Matern or Rational Quadratic (RQ) kernel. Here, we assume that a small amount of additional training data is available for the Matern kernel, whereas no extra training data is provided for the RQ kernel. It is important to note that both the Matern and RQ kernels can be seen as different generalizations of the RBF kernel. Our expectation is that, if the SMCS framework is well trained to transform latent samples from the variational posterior into the true posterior using the Matern kernel data, the improved inference quality will also transfer to the RQ kernel, which is another generalization of the RBF kernel, despite the absence of direct training data. Table 1 shows the expected results, demonstrating that when the training and validation datasets share underlying characteristics, our learned intermediate distributions and transition kernels enable the NP to generalize effectively to the validation dataset.

## 4.2   Varying dimensional tasks with DANP

In this section, we design our experimental setup as follows. Leveraging the ability of the DANP model to handle data with varying input dimensions, we first pre-train the model using data generated from 3D and 4D GP with RBF kernels. Then, we generate training datasets from previously unseen 1D and 2D GP data with RBF kernels to train both the fine-tuned baseline and our TTSNP model. After training, we evaluate both models on 1D GP data generated in two different settings:

Table 2: Results of the context and target log-likelihoods for the GP regression task using the DANP models. "$n$D $A$" in the first column denotes that the $n$-D GP dataset was used to train for method $A$.

| Model | 1D RBF | | Input range shift | | Hyperparameter range shift | |
|---|---|---|---|---|---|---|
| | context | target | context | target | context | target |
| Pre-train | 1.0488 ±0.001 | 0.4188 ±0.002 | 1.0921 ±0.000 | 0.2345 ±0.007 | 0.8136 ±0.000 | -0.1971 ±0.015 |
| SMCS | 1.2725 ±0.001 | 0.4836 ±0.004 | 1.1874 ±0.001 | 0.4226 ±0.003 | **1.2051** ±0.001 | 0.1700 ±0.014 |
| 1D Fine-tune | 1.1863 ±0.005 | 0.5095 ±0.004 | 1.1270 ±0.001 | 0.4663 ±0.004 | 0.8594 ±0.000 | 0.1084 ±0.004 |
| 1D TTSNP (Ours) | 1.2729 ±0.001 | **0.6612** ±0.004 | **1.2639** ±0.001 | **0.6220** ±0.005 | 1.0890 ±0.001 | **0.3615** ±0.001 |
| 2D Fine-tune | 1.0840 ±0.001 | 0.4150 ±0.003 | 1.0859 ±0.001 | 0.3809 ±0.001 | 0.9853 ±0.000 | 0.1485 ±0.001 |
| 2D TTSNP (Ours) | **1.3099** ±0.001 | 0.6026 ±0.003 | 1.2429 ±0.000 | 0.5913 ±0.003 | 1.0414 ±0.000 | 0.3075 ±0.000 |

Table 3: Log-likelihood results for context and target values were obtained for image completion tasks using the EMNIST and Corrupted EMNIST datasets with 3 different corruptions.

| Model | EMNIST | | Corrupted | |
|---|---|---|---|---|
| | context | target | context | target |
| Pre-train | 1.334 ± 0.003 | 0.574 ± 0.003 | 1.241 ± 0.003 | 0.551 ± 0.004 |
| SMCS | 1.349 ± 0.005 | 0.591 ± 0.010 | **1.351** ± 0.002 | 0.573 ± 0.005 |
| Fine-tune | **1.352** ± 0.004 | 0.601 ± 0.004 | **1.351** ± 0.002 | 0.583 ± 0.005 |
| TTSNP (ours) | 1.350 ± 0.002 | **0.620** ± 0.003 | **1.351** ± 0.001 | **0.613** ± 0.004 |

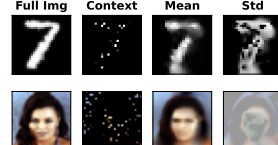

Figure 5: Visualization of the generated mean and covariance from TTSNP for the EMNIST and CelebA image completion tasks. Each column corresponds to (left) the full image, (middle left) the given context, (middle right) the predicted mean, and (right) the predicted standard deviation, respectively.

(1) data within the same distribution as the training set, and (2) data with covariate shifts, such as a shifted input range or modified RBF kernel hyperparameters. This setup allows us to assess model performance both on in-distribution inference and under covariate shift conditions. The results in Table 2 confirm that the learned intermediate distributions and transition kernels remain effective even when the input dimension changes, as long as there is an underlying shared structure between the training and evaluation data—such as being generated from the same RBF kernel, or involving shifts in input range or kernel hyperparameters. This demonstrates the robustness of TTSNP under various covariate shift scenarios.

## 4.3 Image Completion

In the image completion scenario as well, we adopt a setup based on the DANP module. First, leveraging the capability of DANP to handle varying output dimensions, we pre-train the DANP model on the CelebA [37] image completion task. After this pre-training phase, we perform few-shot fine-tuning of the pre-trained DANP model on image completion tasks constructed from the EMNIST [9] dataset. Then we evaluate TTSNP with the EMNIST validation set and the Corrupted EMNIST dataset. Here, we used three different image corruption methods and report the averaged results from these corruptions: 1) 'snow', 2) 'flip', and 3) 'brightness'. Table 3 clearly shows that, compared to other baselines, TTSNP more effectively adapts to new tasks with changing output dimensions, resulting in improved probabilistic inference. This shows that TTSNP is effective not only in scenarios where the input dimension varies but also in cases where the output dimension changes. As long as the task used to train the NP model shares some underlying features with the target task, TTSNP can leverage a small amount of additional data to successfully enhance inference performance. Also, similar to previous experiments, the learned SMCS procedure generalizes well to the unseen shifted evaluation tasks. Fig. 5 shows the predictive mean and standard deviation of EMNIST and CelebA data from TTSNP.

## 4.4 Ablation

**Ablation on the number of pseudo context points.** We conducted an ablation study to investigate the effect of the number of pseudo context points on model performance using the NP model for GP regression with an RBF kernel. Our default configuration employs 12 pseudo representations. To analyze the sensitivity to this design choice, we additionally evaluated models with 3 and 48 pseudo representations. As summarized in Table 4 (left), increasing the number of pseudo representations

Table 4: Ablation studies on (left) the number of pseudo context points and (right) the recalibrated variance in constructing intermediate distributions.

| Number of Representations | 3 | 12 (default) | 48 |
|---|---|---|---|
| Context | 0.880±0.001 | 0.893±0.001 | 0.895±0.001 |
| Target | 0.423±0.002 | 0.430±0.001 | 0.438±0.001 |

| Type of Variance | Learnt | Fixed |
|---|---|---|
| Context | 1.272±0.001 | 1.273±0.001 |
| Target | 0.463±0.003 | 0.661±0.004 |

consistently improves both context and target performance, suggesting that richer pseudo context provides better guidance for latent inference.

**Ablation on the recalibrated variance.** We further performed an ablation on the role of recalibrated variance in constructing intermediate distributions for test-time refinement. The recalibrated variance mitigates the underestimation of uncertainty in the original variational posterior, enabling more accurate and stable inference without additional training. To validate this effect, we compared TTSNP using DANP models on a 1D GP regression task using either the learnt variance or a recalibrated fixed variance. The results, reported in Table 4 (right), demonstrate that recalibrated variance yields improved target prediction performance, highlighting its importance in our refinement strategy.

**Ablation on the number of training data** Here, we evaluate the learning efficiency and convergence speed of TTSNP compared to the Fine-tune baseline by analyzing their target log-likelihood performance. Both methods start from the same pretrained NP model trained on 1D GP data generated with an RBF kernel. We vary the number of training data batches from 10 to 1000 and measure performance using the mean and standard deviation of the target log-likelihood across five random seeds. As shown in Fig. 6, TTSNP achieves significantly faster convergence and consistently outperforms the Fine-tune method, even with substantially fewer training samples. This result shows that our method not only samples more improved latent variables but is also more efficient to train compared to the Fine-tune.

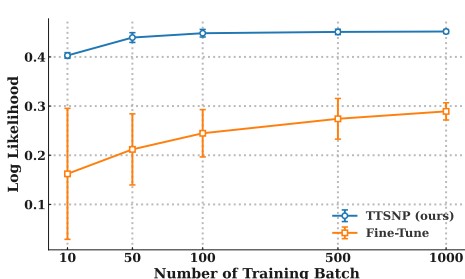

Figure 6: Comparison of target log-likelihood between TTSNP and the baseline 'Fine-tune' under varying amounts of training data, using the same number of samples.

Refer to Appendix D for additional experiments, including ablation studies on the number of samples and training objectives.

## 5 Conclusion

In this paper, we first observed that the latent variational posteriors of existing NP models are often miscalibrated. To address this issue, we proposed TTSNP, a test-time scaling method based on a learned SMCS framework that transforms samples from the variational posterior into ones that better match the true posterior. TTSNP achieves this by learning both the construction of intermediate distributions through pseudo context representations conditioned on the input context and the approximation of score functions required for transition kernels. This enables TTSNP to guide samples more efficiently and effectively toward the true posterior by leveraging diverse and informative intermediate distributions. Our experiments demonstrate that TTSNP significantly improves posterior inference quality across a wide range of settings, including changes in input and output dimensions, covariate shifts, and cross-task generalization scenarios.

Nonetheless, TTSNP comes with some limitations. Training the model requires additional data and introduces extra memory and computational costs. Moreover, unlike standard pre-trained NPs, TTSNP requires additional computation at inference time, such as gradient evaluations for the SMCS procedure. Although this overhead can be mitigated by reducing the number of SMCS steps—allowing for a trade-off between efficiency and sample quality—future work may focus on minimizing this computational burden while maintaining strong performance.

# Acknowledgement

This work was supported by Institute of Information & Communications Technology Planning & Evaluation(IITP) grant funded by the Korea government(MSIT) (No.RS-2019-II190075, Artificial Intelligence Graduate School Program(KAIST)), Institute of Information & Communications Technology Planning & Evaluation(IITP) grant funded by the Korea government(MSIT) (No.RS-2022-II220713, Meta-learning Applicable to Real-world Problems), Institute of Information & Communications Technology Planning & Evaluation(IITP) grant funded by the Korea government(MSIT) (No.RS-2025-02219317, AI Star Fellowship(Kookmin University)), Artificial intelligence industrial convergence cluster development project funded by the Ministry of Science and ICT(MSIT, Korea)&Gwangju Metropolitan City, and the MSIT(Ministry of Science, ICT), Korea, under the Global Research Support Program in the Digital Field program(IITP-2024-RS-2024-00417958) supervised by the IITP(Institute for Information & Communications Technology Planning & Evaluation).

This work was also supported by the Institute of Information & Communications Technology Planning & Evaluation (IITP) with a grant funded by the Ministry of Science and ICT (MSIT) of the Republic of Korea in connection with the Global AI Frontier Lab International Collaborative Research, and by the Samsung Advanced Institute of Technology (under the project Next Generation Deep Learning: From Pattern Recognition to AI).

This work was also supported by the NIH/NHLBI Award R01HL148248, NSF Award 1922658 (NRT-HDR: FUTURE Foundations, Translation, and Responsibility for Data Science), NSF CAREER Award 2145542, ONR N00014-23-1-2634, NSF 2404476, and Apple.

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

# A  Related Works

**Neural Processes**   The original Conditional Neural Process (CNP) [18] introduced a simple encoder-decoder architecture based on multilayer perceptrons. Its extension, the Neural Process (NP) [19], incorporated a global latent variable to model epistemic uncertainty. In this paper, they first suggest the ELBO objective to train the NP models with a latent variable. This design was further improved by Attentive Neural Processes (ANP) [28], which integrated attention mechanisms into the encoder for more expressive context representations. On the other hand, Gordon et al. [21] and Foong et al. [17] incorporate a convolutional deep set module to introduce translation equivariance into NP modules. However, since these models rely on convolutional computations, they are structurally limited and difficult to extend to data beyond three dimensions. Recently, Transformer Neural Processes (TNP) [48] replaced MLPs or attention layers with masked transformer layers to enhance scalability and long-range interaction modeling. Building on TNP, the Dimension-Agnostic Neural Process (DANP) [33] introduced the Dimension Aggregator Block (DAB), enabling a single model to handle tasks with varying input-output dimensions. Our proposed test-time scaling via SMC is developed on top of latent-path-based models such as NP and DANP.

Beyond these, many works have explored improved uncertainty quantification, expressiveness, and inference strategies. For instance, Functional Neural Processes (FNP) [38] used local latent variables to better capture function-specific variations. Bootstrapping Neural Processes (BNP) [34] introduced a residual bootstrapping mechanism to address model misspecification, while Martingale Posterior Neural Processes (MPNP) [32] proposed martingale posterior distribution [16] theory based Bayesian inference for more principled uncertainty estimation. Recent variants like Translation-Equivariant Transformer Neural Processes (TE-TNP) [4], Latent Bottlenecked Attentive Neural Processes (LBANP) [13], and Mixture-of-Experts Neural Processes (MoE-NP) [55] improve generalization and scalability through architectural innovations such as equivariant design, latent bottlenecks, and mixture modeling. Other contributions include Autoregressive Conditional Neural Processes (AR-CNP) [6] for sequential prediction, Self-normalized Importance-weighted Neural Processes (SI-NP) [56] for refined inference, Constant Memory Attentive Neural Processes (CMANP) [14] for memory-efficient modeling, and Gaussian Neural Processes (GNP) [39] for tractable predictive covariance modeling.

**Sequential Monte Carlo**   Our work draws upon foundational concepts in SMC methods and their integration with learned stochastic dynamics. While SMC is designed to efficiently sample from sequential distributions, it has also been extensively explored for sampling complicated posterior distributions through sequential tempering [43], introducing learnable variational distributions [44, 40], or incorporating Langevin dynamics to replace sequential transitions [7]. In parallel, recent works have focused on inference-time scaling in foundation models such as LLMs and diffusion models, leveraging SMC to facilitate this scaling. Zhao et al. [61] treat the autoregressive prediction of LLMs as an SMC process and define a conditional target distribution with a learnable twisted function, enabling the guiding of pretrained LLMs to handle new tasks. In the diffusion models, Kim et al. [29] enhance sample efficiency by using a tempered SMC sampler that balances exploration and exploitation during the sampling process, achieving superior performance in tasks such as aesthetic reward optimization and multi-objective optimization.

# B  Details about SMCS and Modeling Transition kernels

## B.1   Sequential Monte Carlo Samplers

In this section, we briefly review Sequential Monte Carlo Samplers (SMCS) [11], which form the foundational framework for our test-time scaling approach. For a more comprehensive overview, we refer the reader to Del Moral et al. [11], Dai et al. [10].

Let $\pi(x) := \gamma(x)/Z$ denote a target distribution, where the unnormalized density $\gamma(x)$ is known pointwise but the normalization constant $Z$ is unknown. SMCS aims to approximate expectations with respect to $\pi$, such as $\mathbb{E}\pi[f]$ for a test function $f$, while simultaneously providing an estimate of $Z$. Rather than sampling directly from the target $\pi$, SMCS constructs a sequence of intermediate distributions $\{\pi_t\}_{t=0}^{T}$, beginning with a tractable initial distribution $\pi_0$ and gradually transporting it toward the target $\pi_T := \pi$. These intermediate distributions are designed to interpolate between $\pi_0$

---

**Algorithm 1** Multinomial Resampling

---

**Require:** Particles $\{x_t^i\}_{i=1}^N$ at time step $t$ and corresponding importance weights $\{w_t^i\}_{i=1}^N$
**Ensure:** Resampled particles $\{\tilde{x}_t^i\}_{i=1}^N$ and normalized importance weights $\{\tilde{w}_t^i\}_{i=1}^N$.

 1: Compute the normalized importance weights: $\tilde{w}_t^i = \frac{w_t^i}{\sum_{i=1}^N w_t^i}$ for $i \in \{1, \ldots, N\}$
 2: Compute the Effective Sample Size using $\tilde{w}_t^i$s: ESS $= \frac{1}{\sum_{i=1}^N (\tilde{w}_t^i)^2}$
 3: **if** ESS $< 0.3N$ **then**
 4:     **for** $i = 1$ to $N$ **do**
 5:         Sample $j$ from $\{1, \ldots, N\}$ using multinominal distribution with probability $\{\tilde{w}_t^i\}_{i=1}^N$
 6:         Replace particle $x_t^i$ with $\tilde{x}_t^i = x_t^j$
 7:     **end for**
 8:     Replace normalized importance weights $\tilde{w}_t^i$s with $\tilde{w}_t^i = \frac{1}{N}$ for $i \in \{1, \ldots, N\}$
 9: **else**
10:     Keep particles $\{\tilde{x}_t^i\}_{i=1}^N$ and normalized importance weights $\{\tilde{w}_t^i\}_{i=1}^N$
11: **end if**

---

and $\pi$, typically becoming increasingly complex as $t$ increases. Since the intermediate distributions are generally not analytically tractable, SMCS employs sequential importance sampling to adjust the samples accordingly.

In detail, let $\gamma_t$ be the unnormalized density of $\pi_t$, i.e., $\pi_t(x) = \gamma_t(x)/Z_t$ for $t = 0, \ldots, T$ (note that $Z_0 = 1$ as $\pi_0$ is assumed to be fully known). SMCS introduce *forward transition kernels* $\{F_t(x_t \mid x_{t-1})\}_{t=1}^T$ which propagates a sample $x_{t-1}$ to $x_t$. Starting with $x_0 \sim \pi_0$, a sample path $x_{0:T} := (x_0, \ldots, x_T)$ is generated from the joint proposal with density,

$$Q(x_{0:T}) := \pi_0(x_0) \prod_{t=1}^T F_t(x_t \mid x_{t-1}). \tag{18}$$

In principle, one would like to use the marginal $\int Q(x_{0:T}) \mathrm{d}x_{0:T-1}$ as a proposal density for the target $\pi$, but this is generally intractable. Instead, SMCS augments the target to the path space $x_{0:T}$ with a sequence of *backward transition kernels* $\{B_{t-1}(x_{t-1} \mid x_t)\}_{t=1}^T$ and defines the unnormalized path density

$$\Pi(x_{0:T}) := \frac{\Gamma(x_{0:T})}{Z} \text{ where } \Gamma(x_{0:T}) = \gamma(x_T) \prod_{t=1}^T B_{t-1}(x_{t-1}|x_t). \tag{19}$$

A sample path $x_{0:T}$ drawn from $Q$ is then corrected with the importance weights,

$$w_T(x_{0:T}) := \frac{\Gamma(x_{0:T})}{Q(x_{0:T})} = \frac{\gamma(x_T) \prod_{t=1}^T B_{t-1}(x_{t-1} \mid x_t)}{\pi_0(x_0) \prod_{t=1}^T F_t(x_t \mid x_{t-1})}. \tag{20}$$

Given the forward and backward kernels, SMCS starts by drawing $N$ i.i.d. *particles* $\{x_0^i\}_{i=1}^N$ from $\pi_0$. At each iteration, the particles are sequentially extended by drawing samples from the forward kernel $F_t$, and the extended particles are re-weighted through the importance weights, which are computed in recursive fashion as follows:

$$w_0(x_0^i) = 1, \quad w_t(x_{0:t}^i) = w_{t-1}(x_{0:t-1}^i) \times \frac{\gamma_t(x_t^i) B_{t-1}(x_{t-1}^i \mid x_t^i)}{\gamma_{t-1}(x_{t-1}^i) F_t(x_t^i \mid x_t^i)} \text{ for } i = 1, \ldots, N. \tag{21}$$

After completing the sequential update up to the step $T$, we have an estimator for the expectation $\mathbb{E}_\pi[f]$ and the normalization constant $Z$,

$$\mathbb{E}_\pi[f] \approx \frac{1}{N} \sum_{i=1}^N \frac{w_T(x_{0:T}^i)}{\sum_{j=1}^N w_T(x_{0:T}^j)} f(x_T^i), \quad Z \approx \frac{1}{N} \sum_{i=1}^n w_T(x_{0:T}^i). \tag{22}$$

Vanilla SMCS can suffer from particle degeneracy: after several iterations, only a handful of particles carry nearly all the weight while the rest contribute little. A standard remedy is resampling, in which a new particle set is drawn from the current importance-weighted empirical distribution.

In practice, resampling is triggered adaptively, for instance, whenever the effective sample size falls below a chosen threshold. When the resampling is applied, the recursive importance weight updates, and the estimators for expectations and the normalizing constant require minor adjustments. Refer to Algorithm 1 to see how we conducted resampling in this paper. For a full treatment, see Del Moral et al. [11].

**Choice of intermediate distributions.** There are various options for designing intermediate distributions; see Del Moral et al. [11], Dai et al. [10] for examples. A common choice is the geometric annealing path setting $\gamma_t(x) = \pi_0(x)^{1-\beta_t}\gamma(x)^{\beta_t}$ with coefficients $0 = \beta_0 < \beta_1 < \cdots < \beta_T = 1$.

**Choice of transition kernels.** The design of forward and backward kernels is a critical component of SMCS. A widely used approach is to set $F_t$ as a transition invariant to $\gamma_t$, such as Metropolis-Hastings steps, as proposed in Annealed Importance Sampling (AIS) [45]. While there exist optimal backward kernels for given forward kernels [11], they are often intractable in practice. Hence, backward kernels are often chosen for tractability. Notably, in AIS, the forward and backward kernels are coupled in a specific way, allowing simplified importance weight computation that does not even require evaluating the forward transition densities. Alternatively, one may consider proposals based on the Unadjusted Langevin Algorithm (ULA), where the forward transition kernels are defined via a discretization of Langevin diffusion [26, 59, 51],

$$F_t(x_t \mid x_{t-1}) = \mathcal{N}(x_t \mid x_{t-1} + h_t\nabla\log\pi_t(x_{t-1}), 2h_t I), \tag{23}$$

with $h_t$ denoting the step size. When $h_t$ is small, the forward proposal is approximately time-reversible, making the following a natural choice for the backward kernel:

$$B_{t-1}(x_{t-1} \mid x_t) = F_t(x_{t-1} \mid x_t) = \mathcal{N}(x_{t-1} \mid x_t + h_t\nabla\log\pi_t(x_t), 2h_t I). \tag{24}$$

One may also consider underdamped Langevin that incorporates auxiliary momentums [20].

**Choice of resampling threshold** We perform resampling when the effective sample size (ESS) falls below 30% of the total number of particles $N$, i.e., when $\text{ESS} < 0.3N$. This threshold is a conventional choice widely adopted in SMC-based inference methods [53, 7], as it provides a good balance between computational efficiency and sample diversity. We follow the same criterion in our implementation to maintain stability and prevent particle degeneracy during the sequential inference process.

**Learning for SMCS.** Since both the intermediate distributions and the forward/backward kernels are crucial design choices in SMCS, it is beneficial to learn them when possible. Learning for SMCS can be formulated as a variational inference problem, where the ELBO provides a lower bound on the marginal likelihood defined by the SMCS procedure. A common approach is to begin with unadjusted Langevin diffusions and modify the drift term (involving the score functions) to maximize the ELBO [12, 20, 53, 7]. Among these, Chen et al. [7] introduced an optimal control framework for tuning a continuous-time extension of SMCS with resampling, which forms the foundation of our proposed algorithm.

### B.2 Theoretical Justification of the SMC Refinement

While our approach is primarily motivated by empirical observations, its underlying rationale is theoretically supported by established results in the SMC literature. In particular, SMC methods are known to construct a sequence of intermediate distributions that progressively bridge a tractable initial distribution (e.g., a variational posterior) and the true posterior through iterative resampling and transition steps [11, 8]. Under suitable regularity conditions, the weighted particle approximation produced by SMC converges asymptotically to the target posterior as the number of particles increases.

In our setting, initializing SMC with the variational posterior provides a reasonable, though often underdispersed, starting point. The sequential refinement process then mitigates this underdispersion by gradually correcting the approximation mismatch through resampling and transition dynamics. Our method is directly grounded in these established convergence guarantees, providing principled justification for using SMC to enhance posterior quality in latent-variable models.

## B.3 Conceptual Connection to Test-Time Scaling

Test-time scaling (TTS) refers to a family of techniques developed for large language models (LLMs) that enhance inference quality by increasing the computational scale during inference [60]. These approaches include parallel scaling methods such as self-consistency [57] and multi-agent decoding [23], which generate and aggregate multiple reasoning paths, as well as internal scaling strategies like Chain-of-Thought (CoT) prompting [58], which expand the reasoning depth or structure of a single response. Rather than relying on a single greedy decoding trajectory, these methods leverage multiple reasoning traces or extended generation paths to either aggregate or select more plausible outputs, ultimately improving inference reliability and accuracy.

Our proposed approach, TTSNP, builds on the same core idea: increasing computational effort at test time to refine inference. Specifically, this is achieved by using a larger number of SMC steps or by sampling additional latent variables, both of which serve to enhance the quality of posterior approximation. This conceptual parallel motivates the use of the term "scaling" in Test-Time Scaling for Neural Processes, highlighting how our method extends the notion of test-time refinement from LLMs to probabilistic latent-variable models.

## B.4 Model structures

As discussed in § 3.2, our TTSNP framework models both the pseudo representation generator $h$ and the forward/backward transitions $u$ and $v$ using neural transition kernels. In this section, we provide detailed descriptions of how these two neural modules are implemented.

We begin with the pseudo representation generator $h$. As noted in § 3.2, $h$ is designed to be permutation-invariant to ensure exchangeability of the generated context representations. Specifically, it takes the context set representations $R_c$ and a set of random noise variables $\epsilon \sim p(\epsilon)$ (sampled from a standard Gaussian distribution in our experiments) as input and produces randomized pseudo context representations.

Following Lee et al. [32], we implement $h$ using the induced self-attention block (ISAB) [35], a permutation-invariant attention module. The pseudo representation $R_p$ generated by ISAB is defined as:

$$\text{ISAB}(\epsilon) = \text{MAB}(\epsilon, \eta), \quad \text{where} \quad \eta = \text{MAB}(R_c, \epsilon),$$

where MAB denotes Multihead Attention Blocks [54], and $R_c$ is the encoded context representation.

Next, we describe the structure of the drift functions $u$ and $v$, which follow the formulation in Eq. 15:

$$u(\mathbf{r}_t, \tau_t) := \frac{\sigma^2}{2} \left( \nabla \log \tilde{\pi}\_t(\mathbf{r}_t) + (1 - \beta_t) \text{NN}(\mathbf{r}_t, t, \mathcal{D}_c, \mathcal{D}_p) \right), \tag{25}$$

$$v(\mathbf{r}\_t, \tau_t) := -\frac{\sigma^2}{2} \left( \nabla \log \tilde{\pi}_t(\mathbf{r}_t) + (1 - \beta_t) \text{NN}(\mathbf{r}_t, t, \mathcal{D}_c, \mathcal{D}_p) \right), \tag{26}$$

where $\text{NN}(\cdot)$ is a neural network that takes as input the latent state $\mathbf{r}_t$, the time step $t$, and the combined representations of the context and pseudo context sets $\mathcal{D}_c \cup \mathcal{D}_p$, obtained via the pretrained encoder. Including $t$ as input to $\text{NN}$ helps model the adaptive variance of the intermediate posterior $\tilde{q}(\mathbf{r} | \mathcal{D}_c \cup \mathcal{D}_p)$.

The structure of $\text{NN}$ is implemented as a 3-layer Multi-Layer Perceptron (MLP), with the time step $t$ embedded using a sinusoidal positional encoding (PE) as in Vaswani et al. [54]. The final form of the network is:

$$\text{NN}(\mathbf{r}_t, t, \mathcal{D}_c, \mathcal{D}_p) = \text{MLP}\left( \texttt{concat}(\mathbf{r}_t, R_c, R_p) + \text{PE}(t/T) \right),$$

where MLP refers to a 3-layer feedforward network, and $\text{PE}(\cdot)$ is the sinusoidal positional embedding.

## B.5 Details on KL divergence loss

**Path measure and Radon-Nikodym Derivative** The KL divergence loss, $\mathcal{L}_{\text{path}}$, is formulated using two path measures, $\mathbb{P}^u$ and $\mathbb{P}^v$, which correspond to the forward and backward SDEs, respectively. Intuitively, the forward path measure $\mathbb{P}^u$ can be interpreted as the joint distribution $Q(\mathbf{r}_{0:T}^u)$ in the limit as $T \to \infty$ [5]. If this divergence is minimized to zero, it implies that the forward

and backward transitions are perfectly time-reversible, thereby ensuring ideal sampling [7]. Consequently, training the transition kernels with this objective directly enhances the accuracy and quality of samples generated by the SMCS procedure. A key requirement for enabling principled training and inference is the ability to compute the likelihood ratio $w$ between the forward and reverse-time processes, known as the Radon-Nikodym derivative (RND) [53, 7]. This quantity serves as the backbone for defining meaningful objectives and ensuring the correctness of sample-based approximations. Without a tractable form of the RND, learning reliable transition dynamics becomes fundamentally intractable. Fortunately, recent advances [53, 7] provide a computable expression for this crucial quantity as in the following lemma, which we leverage directly in our framework.

**Lemma B.1** (Radon-Nikodym Derivative [53]). *Let $\mathbb{P}^u_{[a,b]}$ and $\mathbb{P}^v_{[a,b]}$ denote the path space measures of the solutions to the forward and backward SDEs defined in Eq. 11 and Eq. 12, respectively, over the time interval $[a, b] \subset [0, 1]$, where the drift functions $u$ and $v$ satisfy Nelson's identity. Assume that the marginal distributions satisfy $\mathbf{r}^u_a \sim \pi_a$ and $\mathbf{s}^v_b \sim \pi_b$. Then, $\mathbb{P}^u_{[a,b]}$-almost surely, the RND between these two measures can be computed as:*

$$\ln w_{[a,b]} = \ln \frac{d\mathbb{P}^v_{[a,b]}}{d\mathbb{P}^u_{[a,b]}}(\mathbf{r}) = \ln \pi_b(\mathbf{r}_b) - \ln \pi_a(\mathbf{r}_a) + \int_a^b \frac{\|u\|^2 - \|u - \sigma^2 \nabla \log \pi\|^2}{2\sigma^2}(\mathbf{r}_t, t) dt$$

$$+ \int_a^b \frac{u - \sigma^2 \nabla \log \pi}{\sigma^2}(\mathbf{r}_t, t) \cdot d\mathbf{r}^v_t - \int_a^b \frac{u}{\sigma^2}(\mathbf{r}_t, t) \cdot d\mathbf{r}^u_t.$$

As we can see in Lemma B.1, we can compute $w$ by divide trajectories $\mathbf{r}^u$ and $\mathbf{s}^v$ into subtrajectories using multiple chunks as follows:

$$\ln w = \ln \frac{\mathbb{P}^v}{\mathbb{P}^u} = \ln \frac{\mathbb{P}^v_{[\tau_0,\tau_1]}}{\mathbb{P}^u_{[\tau_0,\tau_1]}} + \cdots + \ln \frac{\mathbb{P}^v_{[\tau_{T-1},\tau_T]}}{\mathbb{P}^u_{[\tau_{T-1},\tau_T]}} = \ln w_{[\tau_0,\tau_1]} + \cdots + \ln w_{[\tau_{T-1},\tau_T]}. \quad (27)$$

Following Vargas et al. [53] and Chen et al. [7], we can further simplify the continuous-time RND expression from Lemma B.1 by applying a discretization approximation. This yields a practical discrete-time formulation suitable for implementation:

$$\ln w_{[\tau_{n-1},\tau_n]}(\mathbf{r}) = \ln \frac{d\mathbb{P}^v_{[\tau_{n-1},\tau_n]}}{d\mathbb{P}^u_{[\tau_{n-1},\tau_n]}} \quad (28)$$

$$\approx \ln \pi_{\tau_n}(\mathbf{r}_{\tau_n}) - \ln \pi_{\tau_{n-1}}(\mathbf{r}_{\tau_{n-1}}) + \sum_{i=1}^L \frac{\ln p^v_{(n-1)L+i-1}(\mathbf{r}_{((n-1)L+i-1)h}|\mathbf{r}_{((n-1)L+i)h})}{\ln p^u_{(n-1)L+i}(\mathbf{r}_{((n-1)L+i)h}|\mathbf{r}_{((n-1)L+i-1)h})}, \quad (29)$$

where $L$ is the number of steps per subtrajectory and $h$ is the step size. Here, by the Euler-Maruyama discretization, forward and backward transition densities are computed as follows:

$$p^u_{(n-1)L+i-1}(\mathbf{r}_{\text{new}}|\mathbf{r}_{\text{pre}}) = \mathcal{N}(\mathbf{r}_{\text{new}}|\mathbf{r}_{\text{pre}} + hu(\mathbf{r}_{\text{pre}}, a'), h\sigma^2(a')) \quad (30)$$

$$p^v_{(n-1)L+i}(\mathbf{r}_{\text{pre}}|\mathbf{r}_{\text{new}}) = \mathcal{N}(\mathbf{r}_{\text{pre}}|\mathbf{r}_{\text{new}} + h(\sigma^2\nabla \log \pi - u)(\mathbf{r}_{\text{new}}, b'), h\sigma^2(b')) \quad (31)$$

where $a'$ and $b'$ are $((n-1)L + i - 1)h$ and $((n-1)L + i)h$, respectively.

**KL divergence loss**  Then, given the sequential nature of our approach, the KL divergence is evaluated independently on each subinterval $[\tau_{t-1}, \tau_t]$, yielding:

$$\mathcal{L}_{\text{KL}} = \mathbb{E}_{\mathcal{D}_p}\left[\sum_{t=1}^T D_{\text{KL}}\left[\mathbb{P}^u_{[\tau_{t-1},\tau_t]} || \mathbb{P}^v_{[\tau_{t-1},\tau_t]}\right]\right], \quad (32)$$

where KL divergence is averaged by pseudo context due to the randomness $\mathcal{D}_p$. This formulation can be rewritten as a simplified objective using importance weights, following Chen et al. [7]. Following the approaches proposed in Matthews et al. [41] and Chen et al. [7], the KL divergence on each interval can be calculated as:

$$D_{\text{KL}}\left[\mathbb{P}^u_{[\tau_{t-1},\tau_t]} || \mathbb{P}^v_{[\tau_{t-1},\tau_t]}\right] = -\mathbb{E}_{\mathbf{r}^u \sim \mathbb{P}^u_{[\tau_k,\tau_t]}}\left[\log w_{[\tau_{t-1},\tau_t]}(\mathbf{r}^u) \cdot w_{[\tau_k,\tau_t]}(\mathbf{r}^u)\right], \quad (33)$$

---

**Algorithm 2** Overall TTSNP inference algorithm

---
**Require:** Context dataset $\mathcal{D}_c$, trained pseudo context generator $h$, and learned forward and backward transitions $u$ and $v$.
**Ensure:** Updated latent particles $\{\mathbf{r}_T^i\}_{i=1}^N$ and corresponding normalized importance weights $\{\tilde{w}_T^i\}_{i=1}^N$.
 1: **First Stage: Generate pseudo Representation**
 2: Sample noise $\epsilon \sim p(\epsilon)$
 3: Generate pseudo representations $R_p$ using $h(\mathcal{D}_c, \epsilon)$
 4: **Second Stage: Generate initial latent particles and importance weights**
 5: Sample $N$ number of latent particles $\{\mathbf{r}_0^i\}_{i=1}^N$ from variational latent posterior $q(\mathbf{r}|\mathcal{D}_c)$
 6: Set corresponding initial importance weights $\{w_0^i\}_{i=1}^N$ where $w_0^i = \frac{1}{N}$ for $i \in \{1, \ldots, N\}$
 7: **Third Stage: SMCS procedure**
 8: **for** $t = 1$ to $T$ **do**
 9:     **for** $i$ to $N$ **do**
10:         Sample $\mathbf{r}_t^i \sim \mathcal{N}(\mathbf{r}_t^i|\mathbf{r}_{t-1}^i + h_t u(\mathbf{r}_{t-1}^i, \tau_{t-1}), 2h_t I)$
11:     **end for**
12:     Update importance weight based on Eq. 8 and Eq. 29
13:     Resample the latent variables $\mathbf{r}_t^i$ and update $\tilde{w}_t^i$ following Algorithm 1
14: **end for**
15: **Forth Stage: Compute predictive distribution**
16: With final $\{\mathbf{r}_T^i\}_{i=1}^N$ and $\{\tilde{w}_T^i\}_{i=1}^N$ compute the predictive distribution similar to Eq. 38

---

where $\tau_k < \tau_{t-1}$ refers to the most recent resampling time prior to $\tau_{t-1}$. Finally, the KL loss $\mathcal{L}_{\mathrm{KL}}$ can be expressed in practice using importance weights as:

$$\mathcal{L}_{\mathrm{KL}} = \mathbb{E}_{\mathcal{D}_p}\left[\sum_{t=1}^{T}\frac{1}{N}\sum_{i=1}^{N}\mathrm{detach}(w_{\tau_k}^{(i)})\log w_{[\tau_{t-1}, \tau_t]}^{(i)}\right], \tag{34}$$

where $\mathrm{detach}(\cdot)$ indicates that gradients are not propagated through the resampled weights $w_{n-1}^{(i)}$. In our experiment, for efficient training, we trained the model using a single set of $\mathcal{D}_p$.

### B.6 Overall TTSNP latent variable sampling algorithm

In this section, we present the full inference procedure of TTSNP as an algorithm that uses an SMCS procedure to refine latent samples toward the true posterior distribution. As outlined in Algorithm 2, the overall method can be divided into four main stages: (1) 'Generate pseudo representation', (2) 'Generate initial latent particles and importance weights', (3) 'SMCS procedure', and (4) Compute predictive distribution'.

In the first stage, 'Generate pseudo representation', we use the pseudo context generator $h$ to create pseudo representations conditioned on the context set. Next, in 'Generate initial latent particles and importance weights', we sample initial latent variables from the variational posterior $p(\mathbf{r} \mid \mathcal{D}_c)$ and assign initial weights accordingly. These latent particles are then used to initialize the SMCS procedure.

The third stage, 'SMCS procedure', iteratively updates the latent particles using the learned intermediate distributions—constructed using both the context and pseudo representations—and the learned forward and backward transition kernels $u$ and $v$. Finally, in 'Compute predictive distribution', we use the updated latent particles and their associated importance weights to compute the final predictive distribution over target outputs.

## C  Experimental Details

To support reproducibility, we provide our full experimental code as part of the supplementary material. Our implementation is based on the official codebase[1] of DANP [33], and all experiments were performed using PyTorch [3]. Training and evaluation were carried out on either an

---

[1]https://openreview.net/forum?id=uGJxl2odR0

Table 5: Model structure details of NP

| CATEGORY | DETAILS |
|---|---|
| **MODEL SPECIFICATIONS** | |
| HIDDEN DIMENSION FOR DETERMINISTIC PATH | 128 |
| HIDDEN DIMENSION FOR LATENT PATH | 128 |
| MLP DEPTH FOR DETERMINISTIC PATH | 4 |
| MLP DEPTH FOR LATENT ENCODER PRE-MODULE | 4 |
| MLP DEPTH FOR LATENT ENCODER POST-MODULE | 2 |
| DECODER DEPTH | 3 |
| NUMBER OF PARAMETERS FOR 1D GP REGRESSION | 232194 |

Table 6: Model structure details of NP with TTSNP

| CATEGORY | DETAILS |
|---|---|
| **MODEL SPECIFICATIONS** | |
| HIDDEN DIMENSION FOR DETERMINISTIC PATH | 128 |
| HIDDEN DIMENSION FOR LATENT PATH | 128 |
| MLP DEPTH FOR DETERMINISTIC PATH | 4 |
| MLP DEPTH FOR LATENT ENCODER PRE-MODULE | 4 |
| MLP DEPTH FOR LATENT ENCODER POST-MODULE | 2 |
| DECODER DEPTH | 3 |
| NUMBER OF MULTIHEAD ATTENTION BLOCKS IN ISAB | 2 |
| OUTPUT DIMENSION FOR EACH BLOCK | 128 |
| HIDDEN DIMENSION FOR GRADIENT ESTIMATOR | 256 |
| MLP LAYER FOR GRADIENT ESTIMATOR | 3 |
| NUMBER OF PARAMETERS FOR 1D GP REGRESSION | 562818 |

NVIDIA GeForce RTX 3090 or an RTX A6000 GPU. We optimized all models using the Adam optimizer [30], combined with a cosine annealing schedule for the learning rate.

Unless stated otherwise, we selected hyperparameters based on validation log-likelihood across tasks, using the following search spaces: learning rates from $\{5 \times 10^{-5}, 7 \times 10^{-5}, 9 \times 10^{-5}, 1 \times 10^{-4}, 3 \times 10^{-4}, 5 \times 10^{-4}\}$, weight decay values of $\{0, 1 \times 10^{-5}\}$, and batch sizes of $\{16, 32\}$.

## C.1 Model structural details

In this section, we summarize the architectural details of NP, DANP in Table 5 and Table 7, and their respective variants when integrated with the TTSNP method in Table 6 and Table 8. For the NP and NP+TTSNP models, the description is based on the 1D GP regression task. In contrast, both DANP

Table 7: Model structure details of DANP

| CATEGORY | DETAILS |
|---|---|
| MODEL SPECIFICATIONS | |
| HIDDEN DIMENSION FOR LINEAR PROJECTION IN DIMENSION AGNOSTIC BLOCK | 32 |
| HIDDEN DIMENSION FOR SELF-ATTENTION IN DIMENSION AGNOSTIC BLOCK | 32 |
| HIDDEN DIMENSION FOR LATENT PATH TRANSFORMER | 64 |
| NUMBER OF LAYERS FOR LATENT PATH TRANSFORMER | 2 |
| HIDDEN DIMENSION FOR LATENT PATH SELF-ATTENTION | 64 |
| HIDDEN DIMENSION FOR LATENT PATH MLP LAYERS | 128 |
| NUMBER OF LAYERS FOR LATENT PATH MLP LAYERS | 2 |
| HIDDEN DIMENSION FOR DETERMINISTIC PATH TRANSFORMER | 128 |
| NUMBER OF LAYERS FOR DETERMINISTIC PATH TRANSFORMER LAYERS | 6 |
| NUMBER OF HEADS FOR DETERMINISTIC TRANSFORMER LAYERS | 4 |
| DECODER DEPTH | 2 |
| NUMBER OF PARAMETERS | 334562 |

and DANP+TTSNP are designed to maintain a consistent structure and parameter count, regardless of changes in input or output dimensionality.

A key aspect to note is that TTSNP extends the base latent NP architecture by introducing two additional components: a permutation-invariant pseudo representation generator using the ISAB module [35] and a gradient estimator using a Multi-Layer Perceptron (MLP) structure that approximates the score function corresponding to the variational posterior constructed from the pseudo representations.

## C.2 Details on the "Fine-tune"

As outlined in our main experimental setup, we begin with an NP model pre-trained on a large dataset using an approximate ELBO objective. Following pre-training, we observe that the resulting variational posterior often deviates from the true posterior, particularly in few-shot or distribution-shifted settings.

To address this, we introduce a transition kernel and a pseudo representation generator trained using a small amount of additional data. These components refine the latent samples while keeping the pre-trained NP model frozen, ensuring that the core representation knowledge remains intact and enabling robust adaptation to new tasks.

For a fair comparison, the "Fine-tune" baseline uses the same additional data but updates only the latent path of the NP model, with all other components kept frozen. This setup ensures that both methods preserve the pre-trained representation while improving posterior quality through different mechanisms.

Table 8: Model structure details of DANP with TTSNP

| CATEGORY | DETAILS |
|---|---|
| **MODEL SPECIFICATIONS** | |
| HIDDEN DIMENSION FOR LINEAR PROJECTION IN DIMENSION AGNOSTIC BLOCK | 32 |
| HIDDEN DIMENSION FOR SELF-ATTENTION IN DIMENSION AGNOSTIC BLOCK | 32 |
| HIDDEN DIMENSION FOR LATENT PATH TRANSFORMER | 64 |
| NUMBER OF LAYERS FOR LATENT PATH TRANSFORMER | 2 |
| HIDDEN DIMENSION FOR LATENT PATH SELF-ATTENTION | 64 |
| HIDDEN DIMENSION FOR LATENT PATH MLP LAYERS | 128 |
| NUMBER OF LAYERS FOR LATENT PATH MLP LAYERS | 2 |
| HIDDEN DIMENSION FOR DETERMINISTIC PATH TRANSFORMER | 128 |
| NUMBER OF LAYERS FOR DETERMINISTIC PATH TRANSFORMER LAYERS | 6 |
| NUMBER OF HEADS FOR DETERMINISTIC TRANSFORMER LAYERS | 4 |
| DECODER DEPTH | 2 |
| NUMBER OF MULTIHEAD ATTENTION BLOCKS IN ISAB | 2 |
| OUTPUT DIMENSION FOR EACH BLOCK | 64 |
| HIDDEN DIMENSION FOR GRADIENT ESTIMATOR | 192 |
| MLP LAYER FOR GRADIENT ESTIMATOR | 3 |
| NUMBER OF PARAMETERS FOR 1D GP REGRESSION | 475490 |

## C.3 Evaluation Metric for the tasks

Following the approach of Le et al. [31], we evaluate baseline models, such as NP and DANP, using the normalized predictive log-likelihood, which is estimated through Monte Carlo sampling as:

$$\frac{1}{|t|} \sum_{k \in t} \log p(y_{j,k} | x_{j,k}, \mathcal{D}_{j,c}) \approx \frac{1}{|t|} \sum_{k \in t} \log \left( \frac{1}{K} \sum_{k=1}^{K} p(y_{j,k} | x_{j,k}, \mathbf{r}_j^{(k)}) \right), \tag{35}$$

$$\frac{1}{|c|} \sum_{k \in c} \log p(y_{j,k} | x_{j,k}, \mathcal{D}_{j,c}) \approx \frac{1}{|c|} \sum_{k \in c} \log \left( \frac{1}{K} \sum_{k=1}^{K} p(y_{j,k} | x_{j,k}, \mathbf{r}_j^{(k)}) \right), \tag{36}$$

where $\mathbf{r}_j^{(k)}$ are sampled from the variational posterior $q(\theta_j | \mathcal{D}_{j,c})$, and $c$ and $t$ denote the context and target points, respectively. However, for the SMC and TTSNP, we utilize our importance weight $ws$ when computing the predictive log-likelihood as follows:

$$\frac{1}{|t|} \sum_{k \in t} \log p(y_{j,k} | x_{j,k}, \mathcal{D}_{j,c}) \approx \frac{1}{|t|} \sum_{k \in t} \log \left( \sum_{k=1}^{K} \bar{w}_j^{(k)} p(y_{j,k} | x_{j,k}, \mathbf{r}_j^{(k)}) \right), \tag{37}$$

$$\frac{1}{|c|} \sum_{k \in c} \log p(y_{j,k} | x_{j,k}, \mathcal{D}_{j,c}) \approx \frac{1}{|c|} \sum_{k \in c} \log \left( \sum_{k=1}^{K} \bar{w}_j^{(k)} p(y_{j,k} | x_{j,k}, \mathbf{r}_j^{(k)}) \right), \tag{38}$$

where $\bar{w}_j^{(k)}$ are the normalized importance weight for each sample $\mathbf{r}_j^{(k)}$ after the SMC procedures.

## C.4 Dataset details of n-dimensional GP Regression task

To construct GP tasks for our experiments, we generate synthetic datasets using GPs equipped with one of three commonly used kernels: the RBF kernel, the Matern $5/2$ kernel, and the RQ kernel. Each kernel introduces different inductive biases, allowing us to evaluate model robustness across a diverse range of smoothness conditions and correlation structures.

The RBF kernel is defined as

$$k(\mathbf{x}, \mathbf{x}') = s^2 \exp\left(-\frac{\|\mathbf{x} - \mathbf{x}'\|^2}{2\ell^2}\right),$$

while the Matern $5/2$ kernel is given by

$$k(\mathbf{x}, \mathbf{x}') = s^2 \left(1 + \frac{\sqrt{5}d}{\ell} + \frac{5d^2}{3\ell^2}\right), \quad \text{where } d = \|\mathbf{x} - \mathbf{x}'\|.$$

Both use randomly sampled output scales $s \sim \text{Unif}(0.1, 1.0)$ and lengthscales $\ell \sim \text{Unif}(0.1, 0.6)$, and additive Gaussian noise drawn from $p \sim \text{Unif}(0.1, 0.5)$ is applied to the outputs for numerical stability.

The Rational Quadratic kernel, which generalizes the RBF by integrating over a distribution of lengthscales, is defined as

$$k(\mathbf{x}, \mathbf{x}') = s^2 \left(1 + \frac{\|\mathbf{x} - \mathbf{x}'\|^2}{2\alpha\ell^2}\right)^{-\alpha},$$

where we fix the mixture parameter $\alpha = 1.0$. For its implementation, the lengthscale and output scale are sampled similarly from uniform distributions, specifically:

$$\ell = 0.1 + (0.6 - 0.1) \cdot \texttt{Uniform}(0, 1), \quad s = 0.1 + (1.0 - 0.1) \cdot \texttt{Uniform}(0, 1).$$

Given a batch of inputs $x$, the pairwise normalized distances are computed and scaled accordingly, and a small diagonal term $\sigma_\epsilon^2 I$ is added to the covariance matrix to ensure numerical stability.

In our $n$-dimensional GP regression setup, we design the data generation process to scale with input dimensionality, ensuring meaningful predictive inference across varying dimensions.

And for the number of context points $|c|$, we used the sampled number from the range $\text{Unif}(5n^2, 50n^2 - |c|)$ where $n$ indicates the $x$ dimension for the GP task. This quadratic scaling with $n$ reflects the increased data requirements for higher-dimensional input spaces. Similarly, the number of target points $|t|$ is sampled from $\text{Unif}(5n^2, 50n^2 - |c|)$, maintaining a fixed upper limit on the total number of points per task. And, inputs $\mathbf{x} \in \mathbb{R}^n$ for both context and target sets are drawn independently from a uniform distribution over $[-2, 2]^n$.

This flexible GP data generation process allows us to simulate tasks with varying degrees of smoothness, structure, and input dimensionality, providing a rigorous testbed for evaluating uncertainty-aware inference methods.

**Input range shift**     As mentioned earlier, we conducted experiments using inputs $x$ sampled from the range $[-2, 2]$. To evaluate how well TTSNP and baseline methods generalize under covariate shift, we introduced an input range shift scenario. Specifically, we constructed a validation set by modifying only the input domain while keeping all other GP generation settings identical to those used with the RBF kernel. In this shifted setting, inputs $x$ were sampled from $[-3, 1]$, creating a distributional mismatch between training and evaluation. We then assessed the inference performance of each method on this shifted validation set to examine their robustness to covariate shifts in the input space.

**Hyperparameter range shift**     In addition to the input range shift, we also evaluated the generalization ability of TTSNP and baseline methods under a kernel hyperparameter shift, another form of covariate shift where the statistical properties of the data generation process change between training and inference. Specifically, we designed a validation scenario in which only the range of hyperparameters used in the RBF kernel was altered, while keeping other settings—such as the number of data points and the input range—identical to the original training setup.

During training, RBF kernel parameters were sampled from $s \sim \text{Unif}(0.1, 1.0)$ and $\ell \sim \text{Unif}(0.1, 0.6)$. For the shifted validation set, we extended these ranges to $s \sim \text{Unif}(0.1, 2.0)$ and $\ell \sim \text{Unif}(0.1, 1.0)$, effectively increasing the variance and smoothness of the generated functions. We then performed inference on this new validation data to assess how well each method adapts when faced with unseen kernel configurations, thus testing robustness under structural distribution shifts.

## C.5  EMNIST Dataset

For our experiments, we constructed a modified version of the EMNIST Balanced dataset [2] [9], a widely used benchmark derived from the original NIST Special Database. The full dataset includes 47 alphanumeric character classes, but we curated a $10-$class subset for focused evaluation. From this selection, we sampled 24,000 images for training and 4,000 for testing.

Each image is a grayscale digit or letter rendered in a $28 \times 28$ resolution, represented as a single-channel input. Pixel locations were linearly scaled to lie within the range $[-0.5, 0.5]$, and intensity values were normalized to the same interval. During training, for each episode, we randomly chose the number of context points $|c|$ from a uniform distribution over $[5, 45]$, and the number of target points $|t|$ was drawn from $\text{Unif}(5, 50 - |c|)$, ensuring a total of at most 50 points per task.

**Corrupted EMNIST Dataset**  For the Corrupted EMNIST dataset, we evaluated model robustness by applying three types of image corruptions to the EMNIST evaluation set: (1) *snow*, (2) *flip*, and (3) *brightness*.

In the `snow` corruption, we randomly added white noise to the image by setting a subset of pixels to the maximum intensity. Specifically, each pixel was independently selected to be corrupted with probability proportional to the corruption intensity. Formally, a binary mask was generated where each pixel had a probability $0.1\times\texttt{intensity}$ of being set to 1, and the corrupted image was obtained by replacing the masked pixels with maximum brightness.

In the `brightness` corruption setting, we simulated intensity-based corruption by amplifying the pixel values and clipping them to stay within valid bounds. Specifically, we applied the transformation

$$\texttt{data} = \texttt{clamp}(\texttt{data} \times (1 + \texttt{intensity}), \, 0, \, 1),$$

which increases overall brightness proportionally to a given intensity factor, followed by clipping to ensure pixel values remain in the $[0, 1]$ range.

For the `flip` corruption, to simulate geometric distortions, we randomly flipped the image along spatial dimensions. Each image has a $5\%$ chance of being flipped horizontally and another independent $50\%$ chance of being flipped vertically. This introduces structural variations while preserving pixel intensity distributions.

## C.6  CelebA Dataset

For experiments involving natural image completion, we used the CelebA dataset [3] [37], a large-scale face dataset commonly used in generative modeling benchmarks. The dataset consists of 162,770 training images, 19,867 for validation, and 19,962 for testing. All images were center-cropped and downsampled to $32 \times 32$ resolution with 3 RGB channels to reduce computational complexity while preserving key facial features.

As in our EMNIST experiments, we transformed each image into a pixel-level function over spatial coordinates. Specifically, pixel coordinates were scaled to the range $[-0.5, 0.5]$ along both axes, and pixel intensity values in each RGB channel were normalized to the interval $[-0.5, 0.5]$. This setup enables the image completion task to be framed as a conditional regression problem: the model is provided with a subset of known pixel values (the context) and is tasked with predicting the remaining pixels (the targets).

---

[2]https://www.nist.gov/itl/products-and-services/emnist-dataset
[3]https://mmlab.ie.cuhk.edu.hk/projects/CelebA.html

To simulate diverse conditioning scenarios, we sampled the number of context points $|c| \sim$ Unif$(5, 45)$, and drew the number of target points $|t| \sim$ Unif$(5, 50 - |c|)$, ensuring that each training episode contains a variable and realistic number of observed and queried pixels.

# D   Additional experiments

## D.1   Ablation study

**Ablation on training objective**   In this section, we conduct an ablation study on the training objectives used in our method. As described in § 3.3, our full training loss consists of two components: (1) the KL divergence between two path measures, denoted as $\mathcal{L}_{\text{KL}}$, and (2) the log-likelihood maximization loss, denoted as $\mathcal{L}_{\text{LL}}$. To understand the individual contribution of each objective, we evaluate models trained with only one of the two loss terms.

As shown in Table 9, both $\mathcal{L}_{\text{KL}}$ and $\mathcal{L}_{\text{LL}}$ improve sample quality over the baseline variational posterior from the pre-trained

Table 9: Log-likelihood results for the ablation study on training objectives. In the second and third rows, the notation $-A$ indicates that the objective $A$ was excluded from the training loss during model training.

| Model | RBF | |
|---|---|---|
| | context | target |
| TTSNP (ours) | **0.893** $\pm 0.001$ | **0.430** $\pm 0.001$ |
| - $\mathcal{L}_{\text{KL}}$ | 0.880 $\pm 0.001$ | 0.420 $\pm 0.001$ |
| - $\mathcal{L}_{\text{LL}}$ | 0.872 $\pm 0.001$ | 0.415 $\pm 0.001$ |
| SMCS | 0.868 $\pm 0.001$ | 0.405 $\pm 0.001$ |

model, highlighting their effectiveness in enhancing the efficiency and diversity of particles in the learned SMCS procedure. However, the best performance is achieved when both objectives are combined, as in TTSNP. This demonstrates that $\mathcal{L}_{\text{KL}}$ and $\mathcal{L}_{\text{LL}}$ are complementary, working together to improve posterior approximation and overall inference quality.

**Number of samples**   In this subsection, following the setup in § 3.1, we perform an ablation study on the number of samples used during inference. As in other experiments described in § 4, we include Pre-trained, Fine-tune, and SMCS as baselines for comparison. Both TTSNP and SMCS are evaluated with $T = 10$ inference steps.

The experiment is conducted using a pre-trained NP model trained on 1D Gaussian Process data with an RBF kernel. For SMCS, we apply the Unadjusted Langevin Algorithm as the transition kernel to sample from the posterior during inference. The results in Fig. 7 show that TTSNP achieves faster convergence compared to other baselines as the number of samples increases. This demonstrates that the learned intermediate distributions—constructed using pseudo context representations—along with the learned forward and backward transitions, effectively guide latent samples toward the true posterior while capturing diverse modes, resulting in improved sample efficiency.

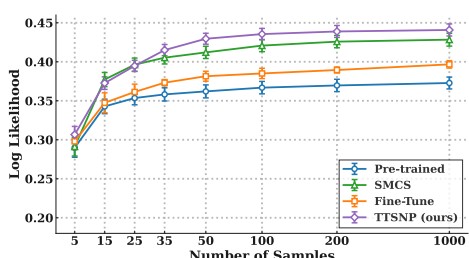

Figure 7: Comparison of target log-likelihood between TTSNP and the baselines 'Pre-trained', 'SMCS', and 'Fine-tune' using NP model trained with GP data generated by RBF kernel. Here, we compare the log-likelihood under the same varying amounts of latent samples.

**Ablation on the Model structure**   In this section, we conduct an ablation study on the architectural design of the drift functions $u$ and $v$. As detailed in Appendix B.4, our default implementation defines the drift functions as follows:

$$u(\mathbf{r}_t, \tau_t) := \frac{\sigma^2}{2} \left( \nabla \log \tilde{\pi}_t(\mathbf{r}_t) + (1 - \beta_t) \, \text{NN}(\mathbf{r}_t, t, \mathcal{D}_c, \mathcal{D}_p) \right), \tag{39}$$

$$v(\mathbf{r}_t, \tau_t) := -\frac{\sigma^2}{2} \left( \nabla \log \tilde{\pi}_t(\mathbf{r}_t) + (1 - \beta_t) \, \text{NN}(\mathbf{r}_t, t, \mathcal{D}_c, \mathcal{D}_p) \right), \tag{40}$$

where $\mathrm{NN}(\cdot)$ is implemented as a 3-layer MLP and is designed to approximate the score function $\nabla \log \tilde{q}(\mathbf{r} \mid \mathcal{D}_c \cup \mathcal{D}_p)$ of the intermediate distribution constructed from the context and pseudo context representations.

While this approach relies on a neural network to approximate the full score function, an alternative formulation can be derived by analytically decomposing the intermediate distribution using the Gaussian score form. In particular, we consider the following decomposition:

$$\nabla \log \pi_t(\mathbf{r}) := \beta_t \nabla \log p(\mathbf{r} \mid \mathcal{D}_c) + (1 - \beta_t) \nabla \log \tilde{q}(\mathbf{r} \mid \mathcal{D}_c) \tag{41}$$

$$+ (1 - \beta_t) \nabla \log \tilde{q}(\mathbf{r} \mid h(\mathcal{D}_c, t, \varepsilon_t) \cup \mathcal{D}_p), \tag{42}$$

$$= \nabla \log \tilde{\pi}_t(\mathbf{r}) + (1 - \beta_t) \frac{\mathbf{r} - \mu_{\mathrm{NN}}(\mathcal{D}_c, \mathcal{D}_p, t)}{\sigma_{\mathrm{NN}}^2(\mathcal{D}_c, \mathcal{D}_p, t)} \tag{43}$$

where $\tilde{\pi}_t(\mathbf{r})$ denotes the part of the intermediate distribution excluding contributions from the pseudo context, and the neural network models the additional gradient signal induced by the pseudo context generator $h$ by modeling the mean $\mu_{\mathrm{NN}}$ and the covariance $\sigma_{\mathrm{NN}}^2$ of the Gaussian score. Also, more simply, we can design as follows:

$$\nabla \log \pi_t(\mathbf{r}) = \nabla \log \tilde{\pi}_t(\mathbf{r}) + (1 - \beta_t) \frac{\mathbf{r} - \mu_{\mathrm{NN}}(\mathcal{D}_c, \mathcal{D}_p, t)}{\sigma^2}, \tag{44}$$

where $\sigma^2$ is the fixed variance, which we used for calibrating variational posteriors.

In this section, we compare the log-likelihood performance of TTSNP against the alternative drift modeling strategies described above, using 1D GP data generated with an RBF kernel. All methods share the same pre-trained NP base model, allowing for a fair evaluation of how different formulations of the drift function impact inference quality. We ensure a fair comparison by using the same training objectives, number of training samples, and number of inference steps across TTSNP and all other model variants—making the drift model structure the only varying factor. As shown in Table 10, while alternative neural network designs also improve sample quality and log-likelihood performance compared to SMCS, TTSNP achieves the most significant improvement, highlighting the effectiveness of its architecture in modeling the score function and guiding posterior refinement.

### D.2 Additional GP regression experiments

In Table 11, we extend the GP regression experiments beyond those presented in § 4.1 and § 4.2 by exploring additional settings. Specifically, we first replicate the experiments originally conducted on 1D GP data in § 4.1, but now using 2D GP data. This allows us to empirically demonstrate that our method remains effective and directly applicable even when the input dimensionality increases. We first extend the **Sample Quality** experiment described in § 4.1 to a more challenging setting using 2D GP data generated with an RBF kernel. The results are reported in the left table of Table 11. While the overall log-likelihood values are lower compared to the 1D case—due to the increased complexity of the 2D input space—the performance trends remain consistent. Specifically, TTSNP continues to improve sample quality and enables efficient inference, confirming its effectiveness beyond simple 1D tasks and into higher-dimensional scenarios.

Table 10: Log-likelihood results for the ablation study on drift model structures. In the second and third rows, the notations 'Mean and Variance' and 'Mean' indicate that the neural network is used to model both the mean and variance of the Gaussian score function, and only the mean, respectively.

| Model | RBF | |
|---|---|---|
| | context | target |
| TTSNP (ours) | **0.893** ±0.001 | **0.430** ±0.001 |
| Mean and Variance | 0.892 ±0.002 | 0.420 ±0.002 |
| Mean | 0.886 ±0.002 | 0.421 ±0.002 |
| SMCS | 0.868 ±0.001 | 0.405 ±0.001 |

We also replicate the **Matern to RQ** experiment from § 4.1 in the 2D setting. The corresponding results are shown in the right table of Table 11. Again, we observe similar performance patterns to those seen in the 1D experiments, demonstrating that TTSNP enhances particle quality even in few-shot adaptation settings and generalizes effectively to unseen tasks. These results highlight the robustness of the learned intermediate distributions and transition kernels, even under high-dimensional covariate shift conditions.

Table 11: Log-likelihood results for Sample Quality (left) and Matern to RQ (right) scenarios.

| Model | RBF | |
|---|---|---|
| | context | target |
| Pre-train | -0.162 ±0.008 | -0.392 ±0.010 |
| Fine-tune | -0.156 ±0.005 | -0.384 ±0.007 |
| SMCS | **-0.154** ±0.004 | -0.367 ±0.004 |
| TTSNP (ours) | -0.156 ±0.004 | **-0.351** ±0.005 |

| Model | Matern | | RQ | |
|---|---|---|---|---|
| | context | target | context | target |
| Pre-train | -0.211 ±0.002 | -0.462 ±0.003 | 0.299 ±0.004 | 0.084 ±0.005 |
| Fine-tune | -0.202 ±0.003 | -0.440 ±0.005 | 0.310 ±0.004 | 0.092 ±0.005 |
| SMCS | **-0.190** ±0.002 | -0.432 ±0.002 | **0.318** ±0.003 | 0.104 ±0.004 |
| TTSNP (ours) | **-0.190** ±0.002 | **-0.420** ±0.001 | 0.316 ±0.003 | **0.110** ±0.003 |

Table 12: Log-likelihood results on the CelebA task. The model was pre-trained on CelebA and evaluated after cross-domain adaptation.

| Model | Context | Target |
|---|---|---|
| Pre-train | 4.148±0.001 | 2.028±0.005 |
| SMCS | 4.150±0.002 | 2.035±0.004 |
| Fine-tune | 4.138±0.003 | 2.019±0.004 |
| TTSNP | 4.150±0.002 | 2.038±0.004 |

## D.3 Additional Image Completion experiment results

**Cross-Domain Generalization and CelebA Evaluation.** Our image completion experiment was designed to evaluate the model's ability to generalize across domains. Specifically, we first pre-trained the DANP model on CelebA and subsequently trained only the transition kernel and pseudo representation generator on EMNIST, while keeping the parameters of the pre-trained base model frozen. This setting isolates the contribution of the SMC refinement procedure, allowing us to assess whether the model can adapt to EMNIST and corrupted EMNIST tasks without retraining the base network. Because the base model remains unchanged, its performance on CelebA under standard variational posterior sampling is expected to stay constant; hence, CelebA was not included in the main evaluation.

Nevertheless, as shown in Figure 5, we include qualitative samples from both CelebA and EMNIST to demonstrate that the model retains its generative capability on the source domain even after adaptation. To further support this observation, we report the quantitative results on the CelebA dataset in Table 12. The results confirm that TTSNP maintains or slightly improves log-likelihood compared to the pre-trained and fine-tuned baselines, indicating strong robustness and generalization.

# E    Broader Impact

This paper identifies a key limitation in latent Neural Process models: after pre-training, the global latent variables tend to be miscalibrated, resulting in low-quality samples from the variational posterior. To address this, we propose a method based on a Sequential Monte Carlo sampler that refines these latent samples to better match the true posterior distribution. While this approach introduces additional computational cost at test time compared to standard inference with pre-trained models, the cost is adjustable based on user preferences and is justified by the significant improvement in probabilistic inference quality. Aside from this computational trade-off, the proposed method poses no known negative societal impacts.

