# OpenReview forum: "Test Time Scaling for Neural Processes"
_NeurIPS.cc/2025/Conference — NeurIPS 2025 poster_

### Official Review · Reviewer_afZz · 2025-06-30

**Clarity:** 2
**Significance:** 3
**Originality:** 3
**Rating:** 5
**Confidence:** 4

**Summary:**

This paper proposes employing SMC samplers to improve the variational posterior in Neural Processes, with applications to meta learning. The authors claim that variational posteriors are typically miscalibrated, providing poor uncertainty quantification.

The authors propose a method that learns forward and backward kernels for the SMC sampler. The kernels are cleverly constructed to have desirable properties by construction, and are trained such that they are approximately time inversions of each other.

The kernels make use of additional synthetic data, which I understood to come from a generative model that is additionally trained.

The authors show experimentally that their method produces superior log-likelihood results.

**Questions:**

- How should we interpret the KL divergence  $KL(q( r | \mathcal{D}_j) || q(r | \mathcal{D}_c ))$? I read this as the divergence between the variational posterior conditioned only on the context in $j$ (the train data of task $\tau_j$) and the variational posterior conditioned on the full data set for task $j$. I think I expected a divergence between some sort of prior and a conditional posterior
- In Equation (2): does it ever make sense to use non diagonal covariance for $y$?
- How is the prior $p(r)$ defined, and how does that relate to the claim of "NPs can capture the data-generating mechanism underlying diverse tasks _without relying on explicit prior assumptions_"?
- Can the authors motivate using the metric of "log-likelihood" to measure the quality of uncertainty quantification?
- What is the intuition for including log-likelihood in the loss for the transition kernels? Shouldn't only using the KL-loss lead to a more accurate posterior?
- How is the model that produces pseudo contexts trained?

**Ethical Concerns:**

["NO or VERY MINOR ethics concerns only"]

**Final Justification:**

My original concerns were mostly in regards to presentation, and some confusion about the results and training procedure. I feel that the authors have adequately answered my questions and addressed the concerns of other reviewers. I am confident that they can incorporate their answers into the main text in time for the camera ready, as well as resolve the minor typos and errors.

**Limitations:**

The authors have adequately discussed the limitations of their work.

**Paper Formatting Concerns:**

No concerns.

**Quality:**

3

**Strengths And Weaknesses:**

The proposed method is novel, and interesting. I also agree with the authors that uncertainty quantification and meta-learning are both very relevant fields of research. I like the motivation of these methods and while the experiments are not particularly extensive I still believe the results are significant enough to establish this method as an interesting area of future research.

Parts of the paper are very clearly written, with a strong story that is simple to follow because the authors often dedicate text to create intuition. I particularly like the background on SMC samplers and the second paragraph of section 3.2

Unfortunately, other aspects of the paper are not so clear, even after consulting the appendix. For example, I do not fully understand how the model that generates pseudo contexts is learned. I also could not find how the prior $p(r)$ is defined.

Some of the captions of Figures are also not very insightful. For example, Figure 3 is impossible to interpret without consulting the main text.

Finally, the experiments do not fully support the claims made by the authors. For example, the paper mentions accurate uncertainty estimation, but most experiments test only for log-likelihood, which to the best of my knowledge tests predictive accuracy (even when tested on ood data.) For example, the authors make strong claims such as "lack of diversity and poor calibration" even though the experiment tests for predictive accuracy.

I think the work is interesting but am leaning to reject this paper due to the clarity issues that I perceive. I would be happy to update my score appropriately if these issues are resolved.

Some minor comments:
- $\tau_j$ seems to by synonymous with $j$, making the use of $\tau$ for tasks unnecessary. Furthermore $\tau$ is also used later for time in SMC, overloading the definition.
- In Lines 73-74 and equations 2: While it is possible to infer the meaning from context, I think the functions $\mu$, $\sigma$ and $q$ could be introduced more properly.
- Line 149: "such as those arising from unseen kernels" -> It is unclear what this means until the reader gets to the experiment section where GPs get mentioned for the first time.
- The sentence in lines 21-24 is very long and difficult to parse
- Line 25: have -> has
- Line 48: I would write plural "samplers" instead of "sampler".
- Line 51: "with following" -> "with the following"

---

> ### Author Rebuttal · Authors · 2025-07-31
>
> > **W1, Q3) How is the prior p(r) defined? How does that relate to the claim of “NPs can capture the data-generating mechanism underlying diverse tasks without relying on explicit prior assumptions”?**
>
> Thank you for your constructive comment regarding the prior distribution $p(r)$. To directly answer your question, when constructing intermediate distributions and the true posterior distribution within our SMC sampler, we adopted a standard Gaussian distribution as the prior. This choice facilitates efficient evaluation of the unnormalized densities required for importance weighting and transition updates during the SMC process.
>
> The statement you referred to, "NPs can capture the data-generating mechanism underlying diverse tasks without relying on explicit prior assumptions," relates closely to our discussion addressing Q1. As explained earlier, Neural Processes are trained to maximize the log-probability of target points rather than the full joint log-likelihood of context and target sets. Thus, the KL divergence in NP training is computed between variational distributions without explicitly modeling a prior distribution.
>
> However, in our framework, because our goal is to transform variational posterior samples toward the true posterior using SMC, evaluating the unnormalized posterior, which includes the prior, becomes necessary. To maintain tractability and ensure minimal informative bias, we utilize a standard Gaussian prior for latent variables. This allows efficient and effective sampling and weighting during the SMC procedure without imposing strong assumptions on the task structure.
>
> > **W1, Q6) How is the model that produces pseudo contexts trained?**
>
> The pseudo context generator is trained jointly with the transition kernel using the same objective, composed of two main terms: the KL divergence between forward and backward path measures, and the marginal log-likelihood on target outputs.
>
> While the KL divergence ensures theoretically consistent latent dynamics, the log-likelihood term specifically guides the pseudo context generator toward producing diverse, task-relevant pseudo representations. By maximizing this log-likelihood, the generator encourages latent samples to capture multiple modes of the true posterior predictive distribution, thus increasing both the diversity and quality of generated samples.
>
> This effect is clearly demonstrated in Figure 3 (left column), which shows that pseudo contexts generated by our approach lead to a broad range of posterior predictions. This diversity is crucial in enabling the transition kernel to refine latent samples toward accurate and uncertainty-aware predictions effectively.
>
> > **W2) Regarding the caption of Figure 3.**
>
> Thank you for your valuable suggestion to improve the readability of the paper. Figure 3 illustrates the following across each column: (left) inference results using latent samples drawn from a variational posterior constructed solely from pseudo representations without access to true context points; (middle) inference results using latent samples refined through TTSNP’s SMCS procedure; and (right) inference results from the Fine-tune baseline. These figures demonstrate how well the generated pseudo representations cover diverse scenarios, and how the latent samples guided by such diversity can lead to better inference results compared to those drawn from the original variational posterior. In the final manuscript, we will follow your recommendation to enhance readability by improving the figure captions and reference quality accordingly.
>
> > **W3, Q4) Log-likelihood as uncertainty estimation?**
>
> Thank you for your insightful question regarding log-likelihood as a metric for uncertainty quantification.
> Log-likelihood is widely recognized as a principled metric in Bayesian Neural Networks (BNNs) and Bayesian inference broadly, as it assesses both predictive accuracy and uncertainty calibration. Unlike deterministic metrics such as RMSE or MAE, log-likelihood evaluates the entire predictive distribution, making it especially suited to uncertainty-aware modeling [1, 2].
>
> Higher log-likelihood indicates that a model assigns greater probability density to observed data, reflecting predictive correctness and well-aligned uncertainty quantification. Theoretically, maximizing log-likelihood corresponds directly to minimizing the KL divergence between the model’s predictive distribution and the true data distribution, providing a statistically grounded evaluation of Bayesian inference quality [3, 4].
>
> In our work, which leverages Sequential Monte Carlo methods to refine posterior inference in latent spaces, log-likelihood explicitly measures whether our refined latent samples produce predictive distributions that better approximate the true posterior predictive. Thus, log-likelihood serves as a rigorous metric to evaluate improvements in uncertainty-aware predictions, aligning directly with the goals of our method.
>
> **References**\
> [1] Kendall. et al. What uncertainties do we need in bayesian deep learning for computer vision?. \
> [2] Lakshminarayanan. et al. Simple and scalable predictive uncertainty estimation using deep ensembles.\
> [3] Bishop. et al. Pattern recognition and machine learning.\
> [4] MacKay. et al. Information theory, inference and learning algorithms.
>
>
> > **W4) Typos and minor comments.**
>
> Thank you for your helpful comments regarding typos, equations, and phrasing. We appreciate your careful reading of the manuscript. We will incorporate all the suggested corrections into the final version to ensure the paper is clearer and more readable. Your feedback will certainly help improve the overall quality and presentation of our work.
>
> > **Q1) Form of the KL divergence?**
>
> Thank you for your insightful question regarding the KL divergence term in the approximate ELBO for Latent Neural Process models.
> Indeed, the objective originally proposed in [1] typically involves a KL divergence between the variational posterior and the prior (as in Eq. 7). However, unlike the standard ELBO, which maximizes the joint likelihood of both context and target points, this modified formulation explicitly aims at improving predictive performance on target points at test time.
>
> This adjustment leads to an intractable conditional prior distribution over latent variables given context points. To handle this intractability, the authors introduce a variational conditional prior as a practical approximation (Eq. 9 of [1]). Consequently, the resulting KL divergence in the training objective becomes a divergence between two variational distributions rather than between a variational posterior and the original prior.
>
> Thus, the divergence between variational distributions naturally arises from approximating the intractable ELBO to specifically enhance predictive performance on target points, highlighting an important and subtle aspect of these training objectives.
>
> **References**\
> [1] Garnelo. et al. Neural processes.
>
> > **Q2) Does it make sense to use non diagonal covariance of y in eq (2)?**
>
> It is certainly possible in principle. However, models like DANP [1] are specifically designed to handle scenarios where both the input $x$ and output $y$ dimensions vary across tasks, enabling flexible inference across a wide range of tasks without requiring task-specific architectures. This flexibility is one of the key advantages of the model.
>
> If one were to model the output variance using a non-diagonal covariance structure, additional design considerations would be required. Specifically, as the dimensionality of $y$ changes across tasks, the number of covariance parameters increases quadratically,  i.e., $O(n^2)$, which introduces computational and regularization challenges. Ensuring that the covariance matrix remains nonsingular and well-conditioned across tasks would also require careful design choices, such as appropriate parameterizations (e.g., low-rank plus diagonal structure) or regularization techniques.
>
> That said, this direction offers a promising path toward more expressive and accurate uncertainty modeling, and we view it as a valuable avenue for future work. Thank you for your insightful comment regarding this potential extension.
>
> **References**\
> [1] Lee. et al. Dimension agnostic neural processes.
>
> > **Q5) Intuition for including log-liklihood in the loss for the transition kernels? Shouldn’t only using the KL-loss lead to a more accurate posterior?**
>
> Thank you for raising this insightful question. You're correct that our transition kernels are trained using both KL divergence and log-likelihood terms, each serving distinct roles.
>
> First, the KL divergence loss ($\mathcal{L}_{\text{KL}}$) between forward ($\mathbb{P}^{u}$) and backward ($\mathbb{P}^{v}$) path distributions ensures that our learned transition dynamics are theoretically consistent and time-reversible, aligned with recent score-based generative modeling approaches [1]. Minimizing this path-wise KL divergence guarantees optimal sampling properties but does not directly optimize for predictive accuracy or uncertainty quantification.
>
> Second, the log-likelihood maximization term complements this by explicitly guiding the latent samples toward task-relevant predictions. Specifically, it encourages the generation of diverse latent samples that lead to accurate and uncertainty-aware posterior predictions.
>
> In summary, the KL divergence loss ensures principled, theoretically sound latent dynamics, while the log-likelihood loss ensures practical effectiveness in predictive modeling. Together, they enable accurate inference with fewer particles.
>
> **References**\
> [1] Chen. et al. Sequential Controlled Langevin Diffusions.

---

> > ### Comment · Reviewer_afZz · 2025-08-04
> >
> > I thank the authors for the insightful comments, resolving most of concerns that I have. I will increase my score accordingly.

---

> > > ### Author Response · Authors · 2025-08-05
> > >
> > > We truly appreciate your encouraging comments and the thoughtful, constructive feedback on our paper. Your suggestions have been particularly insightful and will significantly contribute to improving the quality of our manuscript. We are committed to reflecting them thoroughly in the final version.

---

### Official Review · Reviewer_cBJS · 2025-06-30

**Clarity:** 3
**Significance:** 2
**Originality:** 3
**Rating:** 5
**Confidence:** 3

**Summary:**

The paper presents a novel approach for generating sample from meta-trained neural processes. The presented approach assume a sequential Monte Carlo approach for sampling the latent representation of the process conditioned on the context data. The variational posterior trained over the later representations is utilised as the proposal distribution $\pi_0$ and the true posterior (which is known up to a constant) is used as the target distribution $\pi_T$. The intermediate distribution $\pi_t$ of the SMC is defined as annealed mixture of true posterior, variational posterior (but with variance fixed and set to $\sigma^2=1$) and and variational posterior conditioned on the context data and a pseudo-context, which is essentially an artificial dataset generated by a neural network $h$. In order to train the forward and backward transition kernels needed for SMC, authors leverage a SDE where the forward and backward drift terms depend on an output of a neural network. This neural network is trained so that the marginal forward and backward measures at time $t$ coincide and are equal to $\pi_t$. Authors evaluate the proposed approach on a number of synthetic and real-world problems.

**Questions:**

- For the experiments with a GP, can authors provide the likelihood of the analytical GP posterior?
- In Section 4: Experiments, you write "2) Fine-tune: the latent path is further adapted for inference by training the transition kernel using the available data at test time", this sounds very vague. Can you provide more details on this procedure?
- Can the approach of SMC with neural transition kernels be applied to an arbitrary Bayesian Inference problem?
- Can you address the concerns I have regarding empirical evaluation (listed in the "Weaknesses" section)? For me this is the main issue with the current submission and I am willing to increase my score if these concerns are sufficiently addressed.

**Ethical Concerns:**

["NO or VERY MINOR ethics concerns only"]

**Final Justification:**

In my review I raised a number of concerns regarding empirical evaluation and requested additional ablations. Authors sufficiently answered my questions and provided additional experiments, which resolved my most important concerns. I find the paper interesting and with issues with empirical evaluation resolved, I believe it should be accepted.

**Limitations:**

yes

**Quality:**

2

**Strengths And Weaknesses:**

Strengths:
+ The proposed method gives model the ability to correct itself during test-time, which is a conceptually novel approach that could potentially lead to very robust models
+ The empirical results presented in the paper show the method delivers a consistent improvement over baseline, but I have some concerns about the evaluation, as listed in the "Weaknesses"

Weaknesses:
- Authors seem to be running the same experiments as in the DANP paper[31], but the results they obtain for the baseline model are vastly different, e.g. in Table 3 of DANP paper authors report the performance of 0.969 for EMNIST Target, but the submission in Table 3 reports 0.574 for EMNIST Target. There are no comments on where this significant discrepancy is coming from.
- The method is only applicable to latent neural process. There is also a family of approaches that does not model the variational distribution over latent space, but instead models the distribution over the target variable directly that authors are aware of and cite [17,20,36] but do not compare against them in the evaluation.
- Even though authors run the experiments on CelebA and show the completed images, they do not report numerical values. As such, I am not sure if there is much value in such an experiment, as we do not have any numerical measure of performance.
- I believe a number of important ablations is omitted from the ablation study. For example, is the pseudo context set really necessary? Do we really need to replace the learnt variance of the variational distribution with $\sigma=1$? These seem to be very critical questions that are left unanswered.

---

> ### Author Rebuttal · Authors · 2025-07-31
>
> > **W1) Authors seem to running the same experiments as in the DANP paper, but the results they obtain for the baseline are vastly different.**
>
> Thank you for your thoughtful comment and for your deep interest in our experiments. The difference between our results and those reported in [1] primarily arises from the differences in experimental settings.
>
> In [1], the authors trained the DANP model jointly on both CelebA and EMNIST datasets, aiming to demonstrate the model’s capacity to handle data with varying dimensionality. This joint training exposes the model to both domains during meta-training, naturally resulting in strong performance across both.
>
> In contrast, as detailed in our experimental setup, we pre-train the base model only on CelebA and then train the transition kernel and pseudo representation generator separately on the new dataset, EMNIST. The goal here is to assess robustness to domain shift by evaluating the model’s ability to adapt to unseen data without retraining the full model. Our method refines latent samples from the variational posterior in a post-hoc manner, using only limited additional data.
>
> **References**\
> [1] Lee. et al. Dimension agnostic neural processes.
>
> > **W2) The method is only applicable to latent neural processes.**
>
> Thank you for your question regarding the comparison with Conditional Neural Process (CNP) models. We focus primarily on Latent Neural Process (LNP) models for two main reasons.
>
> First, our experimental setup is designed specifically for LNPs. Our method targets the miscalibration of the variational posterior commonly observed in LNPs trained on large meta-training datasets. To address this, we introduce a test-time SMCS refinement procedure, training only the transition kernel and intermediate distribution networks while keeping the base model frozen. CNP models, lacking a latent path, cannot benefit from this refinement. Adapting a CNP model requires fine-tuning the encoder (and often the decoder), which risks overfitting and catastrophic forgetting, violating the frozen-base assumption that underlies our comparisons.
>
> Second, prior work consistently shows that LNPs outperform CNPs under equivalent architectures. For example, Attentive Neural Processes outperform Conditional Attentive Neural Processes [1], largely due to the ability of latent variables to capture task-level uncertainty and global context, especially in low-data or extrapolation settings.
>
> That said, in response to your request, we conducted additional experiments on the GP regression task including the CNP model, shown in Table R.3. For fair comparison, we fine-tuned both the encoder and decoder of CNP, unlike TTSNP and other baselines where only the latent path or auxiliary modules were trained.
>
> __Table R.3.__  Log likelihood comparison on GP regression task
> |Model | context| target|
> |---------|-----------|---------|
> |pre-train| 0.654±0.008 | 0.150±0.010 |
> |Fine-tune| 0.650±0.005 | 0.178±0.008 |
> |SMCS| **0.757**±0.004 | 0.186±0.006 |
> |TTSNP| 0.753±0.004 | **0.219**±0.006 |
> |CNP| 0.637±0.004 | 0.146±0.005 |
> |CNP_tune| 0.555±0.002 | 0.173±0.004 |
>
> **References**\
> [1] Kim. et al. Attentive neural processes.
>
> > **W3) Numerical values for CelebA experiments.**
>
> To clarify, our image completion experiment was designed to evaluate cross-domain generalization. Specifically, we first pre-trained the DANP model on CelebA and then trained only the transition kernel and pseudo representation generator on EMNIST, without modifying the parameters of the pre-trained base model. The goal of this setting was to assess whether the model could generalize to EMNIST and corrupted EMNIST tasks using only the SMC procedure. Since the base model remains frozen, its performance on CelebA using standard variational posterior samples remains unchanged. For this reason, CelebA performance was not included in the main evaluation.
>
> That said, as shown in Figure 5, we presented samples from both CelebA and EMNIST to visually demonstrate that the model retains its generative ability on the original CelebA domain after adaptation. This was meant to illustrate that the learned components do not degrade performance on the source domain.
>
> We agree that reporting CelebA performance can provide further evidence of robustness. Accordingly, we include the numerical results on CelebA in Table R.4 and will incorporate these quantitative results in the final version of the manuscript to support this point.
>
> __Table R.4__ Log likelihood results for CelebA task
> |Model | context| target|
> |---------|-----------|---------|
> |pre-train|4.148±0.001|2.028±0.005|
> |SMCS|4.150±0.002|2.035±0.004|
> |Fine-tune|4.138±0.003|2.019±0.004|
> |TTSNP|4.150±0.002|2.038±0.004|
>
>
> > **W4-1) Ablation study on pseudo context set.**
>
> Thank you for your suggestion regarding an ablation study on the number of pseudo context points. In response to your comment, we conducted an ablation experiment on the GP regression with RBF kernel using the NP model, specifically examining how the number of pseudo representations affects performance.
>
> Our default setting generates 12 pseudo representations. To evaluate the impact of both increasing and decreasing this number, we additionally tested two alternative configurations in Table R.5: one with 3 pseudo representations and another with 48 pseudo representations. This comparison allowed us to assess how the model's performance varies with the quantity of pseudo context used for guiding latent inference.
>
> __Table R.5__ Ablation on the number of pseudo contexts.
> |Number of Representations | 3  |12(default)| 48    |
> |-----|----|----|----|
> |Context|0.880±0.001|0.893±0.001|0.895±0.001|
> |Target|0.423±0.002|0.430±0.001|0.438±0.001|
>
> The results from this experiment will be included in the final manuscript to provide a more comprehensive understanding of the role of pseudo context size in our method’s effectiveness. Thank you again for raising this important point.
>
> > **W4-2) Ablation study on the replacement of the learnt variance of the variational distribution**
>
> Thank you for your insightful comment regarding the ablation on recalibrated variance.
>
> The recalibrated variance helps address the underestimation of uncertainty in the original variational posterior, allowing for better construction of intermediate distributions without additional learning. This leads to more accurate and stable inference during test-time refinement.
>
> To validate its contribution, we conducted an ablation on the 1D GP regression task using DANP models, comparing performance with and without recalibrated variance in constructing intermediate distributions in Table R.6. This highlights the effectiveness of our variance adjustment strategy.
>
> __TableR.6.__ Ablation study on the variance
> |Type of Variance  |Learnt variance   |Fixed Variance|
> |----|-------|-------|
> |Context |1.272±0.001|1.273±0.001|
> |Target |0.463±0.003|0.661±0.004|
>
> > **Q1) Can authors provide the likelihood of the analytical GP posterior**
>
> In our experiments with NP and DANP models, we used a GP regression setup with an RBF kernel, randomly selecting at least 5 context and 5 target points per batch, with a total cap of 50 points. The test set included 3000 batches of size 16.
>
> We computed the true GP posterior analytically and evaluated log-likelihoods for both context and target points, obtaining averages of 1.3914 and 0.7265, respectively.
>
> To evaluate TTSNP, we applied a DANP model pre-trained on 3D and 4D GP data to the 1D task without retraining. After test-time refinement, it achieved log-likelihoods of 1.2729 (context) and 0.6612 (target). These results show that TTSNP generalizes well to unseen domains and effectively refines the posterior, achieving near-optimal performance without additional training.
>
> > **Q2) Details of “Fine-tune”**
>
> As outlined in our main experimental setup, we begin with an NP model pre-trained on a large dataset using an approximate ELBO objective. Following pre-training, we observe that the resulting variational posterior often deviates from the true posterior, particularly in few-shot or distribution-shifted settings.
>
> To address this, we introduce a transition kernel and a pseudo representation generator trained using a small amount of additional data. These components refine the latent samples while keeping the pre-trained NP model frozen, ensuring that the core representation knowledge remains intact and enabling robust adaptation to new tasks.
>
> For a fair comparison, the "Fine-tune" baseline uses the same additional data but updates only the latent path of the NP model, with all other components kept frozen. This setup ensures that both methods preserve the pre-trained representation while improving posterior quality through different mechanisms.
>
> We agree that clarity on this configuration is important, and we will include a detailed explanation of both the "Fine-tune" baseline and our proposed training procedure in the appendix of the final manuscript.
>
> > **Q3) Can the approach of SMC with neural transition kernels be applied to an arbitrary Bayesian Inference problem?**
>
> Of course, it is possible. If we are performing Bayesian inference and already have access to a variational posterior over parameters or latent variables—whether it be learned through a training procedure or even based directly on the prior distribution—and assuming we can evaluate both the likelihood and the prior, then our method is applicable.
>
> In fact, for an arbitrary Bayesian inference, one could even construct the framework using non-learned transition kernels such as those from the unadjusted Langevin algorithm. However, given sufficient data, one can still expect to construct a more effective and efficient transition kernel by leveraging a neural network. This learned kernel can better adapt to the structure of the posterior distribution and reduce sample inefficiency compared to fixed, hand-crafted transitions.

---

> > ### Comment · Reviewer_cBJS · 2025-08-03
> >
> > Thank you very much for clarifying the points I raised. Given that authors resolved majority of my concerns, I am willing to increase my score to 5 (Accept).
> >
> > I expect authors to include all the additional experiments in the camera ready version and improve clarity of writing.

---

> > > ### Author Response · Authors · 2025-08-04
> > >
> > > Thank you for your positive feedback and constructive discussion regarding our paper. We will incorporate this valuable discussion into the final manuscript to further improve the quality of our work.

---

### Official Review · Reviewer_qq3i · 2025-07-01

**Clarity:** 2
**Significance:** 3
**Originality:** 3
**Rating:** 5
**Confidence:** 3

**Summary:**

The paper introduces a new method to enhance the performance of NPs by refining predictions at inference time using Sequential Monte Carlo and learned proposal transitions. The core idea is to improve posterior inference by resampling the global latent. The method uses generated pseudo-contexts to evaluate likelihoods and an SMC sampler learned at training time.

The authors conduct experiments across a variety of benchmark tasks and show that TTSNP enjoys superior performance and is broadly applicable.

**Questions:**

- I understand that the method is a general framework and can be incorporated with other methods. Have the authors conducted computational complexity comparisons with other methods? Is there a performance trade off for runtime?

- How is resampling handled in the SMC algorithm? Is there adaptive resampling or effective sample size thresholds?

**Ethical Concerns:**

["NO or VERY MINOR ethics concerns only"]

**Final Justification:**

The authors have provided more explanation and justification that made me better understand the contribution of the paper. However, the authors could improve on their clarity by incorporating the explanation they've given in the rebuttal stage to the main paper.

**Limitations:**

In addition to the weaknesses mentioned, have the authors tried their method on large underlying neural networks? What is the highest number of parameters that the experimented NNs have?

**Quality:**

3

**Strengths And Weaknesses:**

Strengths

- The paper is well-motivated; it addresses limitations like posterior underfitting and poor uncertainty calibration. In my understanding, the idea of refining NP posterior predictions at test-time via SMC sampling is novel within the NP literature.

Weaknesses

- It would be nice to have certain analysis on the approximated posterior. Discussions on how the SMC sampler should be used for different tasks (i.e. is convergence guaranteed?) would also supplement the material.

- I am a little bit confused by the implementation of the learned transition kernels. It would benefit from more detail: are the transitions trained jointly with the base NP?

- There is no ablation experiments on the number of pseudo contexts used.

- I hope the authors could include a more comprehensive discussion on the “scalability” claimed in the paper.

---

> ### Author Rebuttal · Authors · 2025-07-31
>
> > **W1) It would be nice to have certain analysis on the approximated posterior.**
>
> While our method is primarily motivated by empirical observations, there is indeed a theoretical foundation supporting the idea that Sequential Monte Carlo (SMC) methods can improve posterior approximation, especially when starting from an initial approximate distribution such as a variational posterior.
>
> According to theoretical results in the literature—particularly those discussed in [1,2]—SMC methods construct a sequence of intermediate distributions that gradually bridge the gap between a tractable initial distribution (e.g., the variational posterior) and the true posterior. By resampling and applying appropriate transition kernels, SMC ensures that samples are progressively guided toward the target distribution. Under certain regularity conditions, it has been shown that the weighted particle approximation produced by SMC converges to the true posterior as the number of particles and intermediate steps increases.
>
> In our case, the use of the variational posterior as the proposal for the initial distribution means that we start from a reasonable, albeit underdispersed, approximation. The SMC framework then allows us to refine this approximation by gradually correcting the mismatch through resampling and transition dynamics, effectively mitigating the underfitting issue. Therefore, while we do not derive a new theoretical result specific to our method, our approach is directly supported by established results in the SMC literature, and the guarantee of asymptotic convergence justifies its use for improving posterior quality in practice.
>
> **References**\
> [1] Moral. et al. Sequential Monte Carlo samplers.\
> [2] Chopin. et al. Central limit theorem for sequential Monte Carlo methods and its application to Bayesian inference.
>
> > **W2) Implementation of the learned transition kernels. Are the transitions trained jointly with the base NP?**
>
> To clarify, the transition kernels are trained separately from the base Neural Process model. Specifically, our experimental setting assumes that an NP model has already been pre-trained using an approximate ELBO objective. In practice, we empirically observed that the variational posterior of the pre-trained model underfits the true posterior distribution.
>
> To address this, we introduce two additional components, namely the transition kernel and the pseudo representation generator, which are trained using only a small amount of additional data. Importantly, the parameters of the original NP model remain frozen throughout this training procedure. This ensures that performance improvements arise solely from refined posterior inference rather than additional training of the base model.
>
> For a fair evaluation of our method, we also include a baseline called Fine-tune, where the same additional data is used to directly update the latent variational posterior through standard gradient-based fine-tuning. This comparison demonstrates the effectiveness of our approach in improving posterior quality without requiring full retraining of the underlying NP model.
>
> > **W3) Ablation on the number of pseudo contexts used.**
>
> Thank you for your suggestion regarding an ablation study on the number of pseudo context points. In response to your comment, we conducted an ablation experiment on the GP regression with RBF kernel using the NP model, specifically examining how the number of pseudo representations affects performance. We use 12 pseudo representations by default and additionally evaluate configurations with 3 and 48. This comparison allowed us to assess how the model's performance varies with the quantity of pseudo context used for guiding latent inference.
>
> __Table R.1__ Ablation on the number of pseudo contexts.
> |Number of Representations | 3  |12(default)| 48    |
> |-----|----|----|----|
> |Context Log-likelihood |0.880±0.001|0.893±0.001|0.895±0.001|
> |Target Log-likelihood |0.423±0.002|0.430±0.001|0.438±0.001|
>
> The results from this experiment will be included in the final manuscript to provide a more comprehensive understanding of the role of pseudo context size in our method’s effectiveness. Thank you again for raising this important point.
>
>
> > **W4) More comprehensive discussion on the scalability claimed in the paper.**
>
> Thank you for raising such a thoughtful question regarding the core of our work. *Test-time scaling* [1] refers to a class of methods in Large Language Models that enhance inference quality by increasing the *computational scale* at inference time. This includes **parallel scaling** techniques such as *self-consistency* [2] and *multi-agent decoding* [3], where multiple response paths are generated and aggregated, as well as **internal scaling** strategies like *Chain-of-Thought* prompting [4], where the depth or structure of a single response is expanded to improve reasoning.
>
> These methods go beyond the standard greedy decoding of a single response path. Instead, they leverage multiple reasoning traces or longer generation paths to either combine multiple outputs or select the most plausible one—ultimately leading to improved inference quality.
>
> Our approach, TTSNP, shares this underlying philosophy: it increases computational effort at test time—either by using a larger number of SMC steps (T) or by sampling more latent variables—to refine inference. This refinement improves the quality of statistical inference similarly to how test-time scaling improves LLM output. This conceptual parallel is precisely why we adopt the term “*scaling*” in *Test Time Scaling for Neural Processes*.
>
> **References**\
> [1] Zhang. et al. A Survey on Test-Time Scaling in Large Language Models: What, How, Where, and How Well?.\
> [2] Wang. et al. Self-consistency improves chain of thought reasoning in language models. \
> [3] Guo. et al. Large language model based multi-agents: A survey of progress and challenges.\
> [4] Wei. et al. Chain-of-thought prompting elicits reasoning in large language models.
>
>
> > **Q1) Computational complexity comparisons with other methods? Performance trade-off for runtime?**
>
> As discussed in the Conclusion and in our response to W4, TTSNP is fundamentally a method that improves inference quality via computational scaling at test time. Consequently, it naturally incurs a higher computational cost compared to baseline methods that directly sample latent variables from the variational posterior and perform inference without refinement.
> To quantify this increase in cost, we measured the actual computational overhead using the DANP model on the GP Regression task with an RTX 3090 GPU. The results, compared against the Fine-tune baseline under the same settings, are reported as follows:
>
> __Table R.2__ Time complexity comparison for 200 training steps.
> |   Method  | Fine-tune |  TTSNP|
> |------|-----|------|
> |Time (sec) |20 |80 |
>
> Additionally, the relationship between test-time computational cost and inference quality is clearly visualized in Figure 4. As shown in the figure, increasing the number of SMC steps leads to a noticeable improvement in the quality of the latent variable particles. These particles become better aligned with the true posterior distribution, resulting in more accurate inference. Moreover, as the particle distribution converges more closely to the true posterior, we also observe a reduction in the variance of the log-likelihood, which indicates more stable and reliable predictions.
>
> This empirical evidence supports the idea that TTSNP offers a favorable trade-off between computational cost and predictive performance, and that the degree of refinement can be flexibly controlled based on resource availability at test time.
>
> > **Q2) How is resampling handled in the SMC algorithm? Is there adaptive resampling or effective sample size thresholds?**
>
> Thank you for your insightful question regarding the resampling step in our SMC sampler. As you rightly pointed out, this aspect was clearly described in Algorithm 1 in the appendix, but we acknowledge that it was not explicitly stated in the main text.
>
> To clarify, we perform resampling when the effective sample size (ESS) falls below 30% of the total number of particles NN, i.e., when $ESS<0.3N$. This threshold is a conventional choice commonly used in SMC-based methods [1,2], and we adopted the same criterion in our implementation to ensure stability and sample diversity during the inference process.
>
> We appreciate your suggestion and will make sure to incorporate this detail more clearly in the main body of the final manuscript, so that there is no ambiguity regarding the method design.
>
> **References**\
> [1] Vargas et al. Transport meets variational inference: Controlled Monte Carlo diffusions.\
> [2] Chen. Sequential Controlled Langevin Diffusions.
>
> > **L1) Highest number of parameters that the experimented NNs have?**
>
> Thank you for your question regarding the scale of the underlying neural networks used in our experiments. We appreciate your interest in the model capacity and its impact on our method. In our experiments, the largest model we used contains approximately 1 million parameters, which is the largest parameter scale explored in the NP literature, consistent with the setting used in [1]. Also most of the experiments were conducted using the DANP model, which represents the most advanced architectural variant of Neural Processes to date.
>
> Our choice of this model and scale was intentional, as it allows us to evaluate the proposed method under a realistically challenging and expressive model, thereby demonstrating its applicability beyond small-scale or toy settings. While scaling to even larger architectures remains a promising direction, we believe that working with the largest and most expressive NP model to date provides strong empirical evidence of the effectiveness of our method within this research domain.
>
> **References**\
> [1] Lee et al. Dimension agnostic neural processes.

---

> > ### Comment · Reviewer_qq3i · 2025-08-08
> >
> > I thank the authors for their explanations and newly conducted experiments--they have resolved most of the concerns I have. I will raise my score to reflect so.

---

> > > ### Author Response · Authors · 2025-08-08
> > >
> > > Thank you very much for your positive comments and insightful questions. We will incorporate the additional experiments and discussions we conducted to improve the quality of our work and ensure they are well reflected in the final manuscript.

---

### Official Review · Reviewer_1HSn · 2025-07-01

**Clarity:** 2
**Significance:** 2
**Originality:** 2
**Rating:** 4
**Confidence:** 5

**Summary:**

The authors identify an issue with NP: during inference, the learned posterior often deviates from the true posterior, especially at test time. This stems from the quality of the latent samples. Their proposal to address this issue is based on an SMC sampler operating in the latent space. This requires constructing a sequence of distributions, from initial to target. As a result, their proposal, termed TTSNP, "transforms samples drawn from the variational posterior to be better aligned with the true posterior." The proposal is evaluated empirically.

**Questions:**

Please see my questions above

**Ethical Concerns:**

["NO or VERY MINOR ethics concerns only"]

**Final Justification:**

Thanks for the thorough response. I appreciate the clarification on the points raised in my review, in particular the explanation about the appropriateness of the MCMC method considered and the experiments. After reading the rebuttal to my review (and that of other reviewers), I am increasing my score.

**Limitations:**

Partially covered.

To improve the paper, the authors could also comment on the practical application of the method, as the current experiments do not seem to reveal the complete potential of the developed methodology.

**Quality:**

2

**Strengths And Weaknesses:**

The paper is relatively clear, the idea is interesting, and the authors provide experimental evidence of the method's features. However, I fail to recognise the impact of the method, or what "practical" challenge (beyond the posterior mismatch problem mentioned) it addresses that no other method addresses in a better way. There are also some formatting/presentation issues in the manuscript.

I give a more detailed list of points of improvement/questions:

- Fig 3: What's the difference among all plots? This only seems to be explained in lines 270-276 and not in the fig's caption
- It is unclear what the purpose of the experiments is. For instance, the proposed method is tested on image completion (the only experiment with real-world data). However, there are many other task-specific methods for image completion that -- according the results in fig 5 right --- could provide better results
- The method is only tested empirically, mainly with synthetic data. Is it possible to provide any theoretical guarantee that the introduced SMC sampler in the latent space, improved the approximate posterior?
- Captions are not properly produced: fig3 does not state what each plot is, fig2 is not referenced in the text, fig5 lacks "left" and "right" captions.
- Why is the method referred to as "scaling" - what is "scaled" here? The method is said to "sample more efficiently and effectively toward the true posterior", but why is that to _scale_?

---

> ### Author Rebuttal · Authors · 2025-07-31
>
> > **W1) Fail to recognise the impact of the method, or what “practical” challenge (beyond the posterior mismatch problem mentioned) it addresses that no other method addresses in a better way.**
>
> Thank you for your insightful question. As you pointed out, the main contribution of TTSNP lies in addressing the discrepancy between the variational posterior and the true posterior of the latent distribution by employing Sequential Monte Carlo sampler at test time. This approach of refining variational posterior samples—originally underfitting the true posterior—through an exact sampling method to ensure better distribution is the first of its kind in the field of Neural Processes. As also acknowledged by other reviewers (qq3i, cBJS, afZz), this is an important and valuable direction for improving the quality and reliability of predictions when using NPs for statistical inference. Therefore, enhancing latent posterior sampling in neural processes models via TTSNP constitutes a significant methodological advance, enabling better inference in various practical regression and Bayesian optimization tasks.
>
>
> > **Q1) The purpose of the experiments is unclear. However, there are many other task-specific methods for image completion that could provide better results.**
>
> We appreciate the reviewer’s attention to the rationale behind our experiments. The image completion experiment is designed to highlight the capability of NPs as uncertainty-aware meta-learning regression models to predict unseen parts of test images even with only a few given context points—sometimes fewer than 50 pixels in a 32×32×3 image. This task is a standard benchmark for evaluating NPs.
>
> In our experiment, we demonstrate that, unlike other baselines, our model can successfully perform image completion on a target domain that was not meta-learned during the pre-training phase, by training only the transition kernel and the pseudo representation generator. Furthermore, we show that our method remains robust even when the context data is corrupted.
>
> While there certainly exist state-of-the-art methods and model architectures specialized for image completion tasks, our goal is not to outperform such methods. Rather, we aim to show that, similar to the GP regression case, refining latent samples using our method leads to clear performance improvement—underscoring the broader utility of our approach across different tasks that can be addressed by Neural Processes.
>
> > **Q2) Is it possible to provide any theoretical guarantee that the introduced SMC sampler in the latent space improves the approximate posterior?**
>
> Thank you for your insightful and theoretically grounded question regarding the potential guarantees provided by the SMC sampler for improving the approximate posterior in the latent space. While our method is primarily motivated by empirical observations, there is indeed a theoretical foundation supporting the idea that **Sequential Monte Carlo (SMC) methods can improve posterior approximation**, especially when starting from an initial approximate distribution such as a variational posterior.
>
> According to theoretical results in the literature—particularly those discussed in [1,2]—SMC methods construct a sequence of intermediate distributions that gradually bridge the gap between a tractable initial distribution (e.g., the variational posterior) and the true posterior. By resampling and applying appropriate transition kernels, SMC ensures that samples are progressively guided toward the target distribution. Under certain regularity conditions, it has been shown that the **weighted particle approximation produced by SMC converges to the true posterior** as the number of particles.
>
> In our case, the use of the variational posterior as the proposal for the initial distribution means that we start from a reasonable, albeit underdispersed, approximation. The SMC framework then allows us to refine this approximation by gradually correcting the mismatch through resampling and transition dynamics, effectively mitigating the underfitting issue. Therefore, while we do not derive a new theoretical result specific to our method, our approach is directly supported by established results in the SMC literature, and the **guarantee of asymptotic convergence** justifies its use for improving posterior quality in practice.
>
> **References** \
> [1] Moral. et al. Sequential Monte Carlo samplers.\
> [2] Chopin. Central limit theorem for sequential Monte Carlo methods and its application to Bayesian inference.
>
> > **Q3) Why is the method referred to as “scaling” - what is scaled here?**
>
> Thank you for raising such a thoughtful question regarding the core of our work. *Test-time scaling* [1] refers to a class of methods in Large Language Models that enhance inference quality by increasing the *computational scale* at inference time. This includes **parallel scaling** techniques such as *self-consistency* [2] and *multi-agent decoding* [3], where multiple response paths are generated and aggregated, as well as **internal scaling** strategies like *Chain-of-Thought* prompting [4], where the depth or structure of a single response is expanded to improve reasoning.
>
> These methods go beyond the standard greedy decoding of a single response path. Instead, they leverage multiple reasoning traces or longer generation paths to either combine multiple outputs or select the most plausible one—ultimately leading to improved inference quality.
>
> Our approach, TTSNP, shares this underlying philosophy: it increases computational effort at test time—either by using a larger number of SMC steps (T) or by sampling more latent variables—to refine inference. This refinement improves the quality of statistical inference similarly to how test-time scaling improves LLM output. This conceptual parallel is precisely why we adopt the term “*scaling*” in *Test Time Scaling for Neural Processes*.
>
> **References**\
> [1] Zhang. et al. A Survey on Test-Time Scaling in Large Language Models: What, How, Where, and How Well?.\
> [2] Wang. et al. Self-consistency improves chain of thought reasoning in language models. \
> [3] Guo. et al. Large language model based multi-agents: A survey of progress and challenges.\
> [4] Wei. et al. Chain-of-thought prompting elicits reasoning in large language models.
>
> > **Q4) Formatting issues regarding figure captions.**
>
> Thank you for your valuable suggestion to improve the readability of the paper. First, we hereby provide additional information of each columns in Figure 3: (left) inference results using latent samples drawn from a variational posterior constructed solely from pseudo representations without access to the true context points; (middle) inference results using latent samples refined via  TTSNP’s SMCS procedure; and (right) inference results from the Fine-tune baseline. These figures demonstrate how well the generated pseudo representations cover diverse scenarios, and how the latent samples guided by such diversity can lead to better inference results compared to those drawn from the original variational posterior. Secondly, we will make sure to mention Figure 2, which is a schematic of our method, in Section 3. Lastly, we acknowledge that presenting Figure 5 in the same row as Table 3 may cause confusion; we will separate them in the final manuscript to improve clarity. In the final manuscript, we will follow your recommendation to enhance readability by improving the figure captions and reference quality accordingly.

---

> > ### Comment · Reviewer_1HSn · 2025-08-04
> >
> > Thanks for the thorough response. I appreciate the clarification on the points raised in my review. After reading the rebuttal to my review (and that of other reviewers), I am increasing my score.

---

> > > ### Author Response · Authors · 2025-08-05
> > >
> > > Thank you for your positive feedback and constructive discussion on our paper. The suggestions you provided were valuable contributions that will help enhance the quality of our work, and we will make sure to carefully incorporate them into the final manuscript.

---

### Note · Authors · 2025-08-12

Dear Area Chair and Reviewers,

We sincerely thank you for your time, effort, and constructive engagement throughout the review process.
We are delighted that all reviewers found our rebuttals and additional experiments convincing, and that every reviewer raised their score.

Your feedback helped us make several major additions that further strengthened our core claims. Specifically, we:
- Performed new experiments, including an ablation on the number of pseudo contexts used, a time complexity analysis, adding Conditional Neural Process baselines on GP regression tasks, an ablation study on replacing the learnt variance of the variational distribution, and new experimental results on the CelebA dataset.
- Provided a further theoretical discussion on how the SMC sampler can improve posterior approximation when starting from an initial approximate distribution such as a latent variational posterior.
- Addressed formatting and clarity issues, such as improving figure captions and giving a clearer explanation of the term "scaling," which will be included in the final manuscript.

These additions directly address the reviewers' questions on scalability, robustness, clarity, and theoretical justification, as reflected in the unanimous score increases.
We will ensure that all additional experiments, theoretical explanations, and writing improvements are fully incorporated into the final version.
This process has not only reinforced TTSNP as the first principled framework to apply test-time scaling for posterior refinement in Neural Processes, but also improved the overall clarity and readability of the paper.
We are sincerely grateful for the reviewers' insightful feedback and for the opportunity to refine our work into a stronger and clearer contribution.

Sincerely,
The Authors of Paper 18791

---

### Decision · Program_Chairs · 2025-09-17

**Decision:**

Accept (poster)

**Comment:**

This paper presents a neat take on the currently hot topic of "test time scaling" - i.e. spending more computation on model prediction to refine or improve samples from a model - applied to improve predictions from Neural Processes.  They develop a Sequential Monte Carlo sampler to refine initial samples from the variational posterior of Neural Processes towards samples from the true posterior.

The paper received 3 accepts and one borderline accept.  The reviewers initially seemed to find the presentation a bit confusing and in particular asked a variety of clarifying questions about the experiments.  However, the author's clarifications in the discussion seem to have been quite convincing as every reviewer ended up raising their score.  The reviewers found the paper novel, well written and the method compelling.  After discussion, they also found the experiments convincing.   Therefore, the recommendation is to accept the paper.